# Bootstrapping Visual Assistant Modeling with Situated Interaction Simulation

**Yichi Zhang**[1]**, Run Peng**[1]**, Lingyun Wu**[1]**, Yinpei Dai**[1]**,**
**Xuweiyi Chen**[2]**, Qiaozi Gao**[3]**, Joyce Chai**[1]
[1]University of Michigan, [2]University of Virginia, [3]Amazon
{zhangyic,roihn,chaijy}@umich.edu

## Abstract

Visual assistants that can guide humans through complex tasks in physical environments have significant potential, yet their development is hindered by the high cost of human-in-the-loop data collection. We present BA-SIS (Bootstrapping Assistant modeling with Situated Interaction Simulation), a novel framework that fundamentally rethinks how visual assistants are developed and evaluated. Rather than relying on expensive human data collection, BASIS leverages simulation to bootstrap capable assistants through three interconnected stages: (1) Situated Interaction Simulation generates high-quality synthetic data through interactions between oracle assistants and simulated users; (2) Autonomous Model Development trains and continuously evaluates assistant models using this synthetic data; and (3) Real-User Validation verifies effectiveness with human users. We implement BASIS in Alexa Arena and demonstrate that our best model—despite being fine-tuned solely on synthetic data and operating under realistic perception conditions—enables real human users to achieve a 72.9% success rate, approaching the 88.6% performance of an oracle assistant with access to privileged information of perfect perception. Through detailed error analysis, we identify object identification as the primary bottleneck for current visual assistants. Our approach bridges the gap between interaction simulation and real human-AI collaboration, establishing a scalable pipeline for developing assistants that can effectively guide users through complex tasks. Project website: https://colm-basis.github.io/

## 1 Introduction

Visual assistants (Nazim et al., 2022; Waisberg et al., 2024; Plizzari et al., 2024) are an emerging direction of AI systems designed to work alongside humans by understanding their goals, observing their actions, and providing real-time guidance in physical environments. Enabled by wearable devices such as smart glasses, they perceive the world from the user's egocentric view and offer context-aware language support. As illustrated in Figure 1, these assistants operate in a unique interaction paradigm where the AI and human share the same physical space but possess asymmetrical knowledge and capabilities. The assistant leverages task-specific knowledge to interpret human actions and communicates actionable guidance, while the human exercises physical agency to complete the task at hand.

Despite significant advances in AI, developing effective visual assistants faces many challenges. The first challenge comes from the lack of diverse interaction data. Traditional approaches rely heavily on wizard-of-oz methods where human experts act as assistants during data collection (Bao et al., 2023; Wang et al., 2023), making the process prohibitively expensive and limiting the variety of communicative dynamics that may occur during the interaction. Furthermore, previous work typically focuses on benchmarking with static video data (Huang et al., 2024; Grauman et al., 2022; Sener et al., 2022). It does not address interactive evaluation with real-time user feedback, which is difficult but essential for measuring the effectiveness of the assistant.

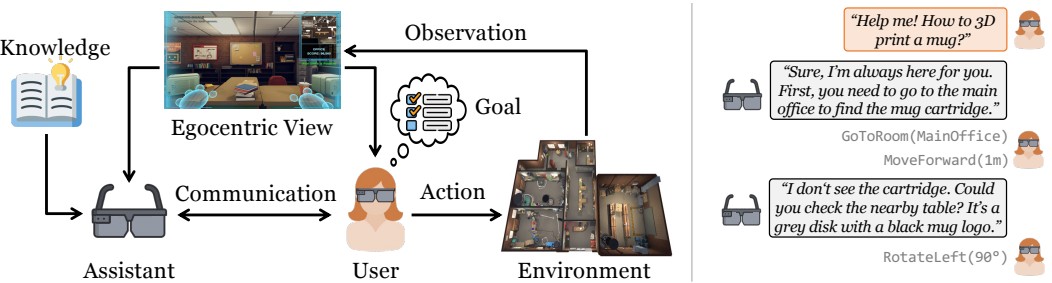

Figure 1: **(Left)** Visual assistant interaction framework: The assistant and user share the same egocentric view but have complementary capabilities. The assistant draws on task knowledge to provide guidance through language communication, while the user follows the guidance and physically acts in the environment to achieve a specific goal. **(Right)** Example situated interactions where the assistant proactively offers guidance.

To address these challenges, we present BASIS (Bootstrapping Assistant modeling with Situated Interaction Simulation), a novel framework that rethinks the visual assistant development cycle by leveraging simulation to replace labor-intensive human data collection. The core idea is to construct highly-realistic, training-free *simulated users* and *oracle assistants* using powerful multimodal large language models (MLLMs) (Clark et al., 2023; Georgiev et al., 2024) for role-playing and interaction. BASIS progresses through three interconnected stages: (1) **Situated Interaction Simulation**: in this stage, MLLM-powered oracle assistants interact with simulated users in a virtual environment to generate contextually grounded interaction trajectories. The oracle assistant has access to privileged information (e.g., ground-truth object positions and labels) to enable accurate language guidance generation. In contrast, simulated users are intentionally designed with asymmetrical, incomplete knowledge to mimic real-world user characteristics, such as ambiguous intent and suboptimal behavior, capturing the complexity of human-assistant interactions. This stage lays the foundation for training assistant models in the next phase. (2) **Autonomous Model Development**: Assistant models are trained using the synthetic interaction data from Stage 1 under more realistic perceptual constraints (i.e., replying on visual observations and no access to privileged information). The assistant's perception is aligned with the user's egocentric view. The simulated users from Stage 1 are also reused to automatically evaluate assistant performance, enabling rapid, low-cost iteration. This stage is crucial for developing and refining assistant models at scale with minimal human intervention. (3) **Real-User Validation**: the refined assistant models are evaluated with real human users to validate that the skills learned in simulation transfer effectively to real-world scenarios. This stage ensures practical usability and grounds simulation-driven development in actual user needs and expectations.

We implement and evaluate BASIS within Alexa Arena, a rich 3D simulator supporting complex object interactions and diverse tasks (Gao et al., 2023). Our experiments demonstrate the framework's effectiveness: a visual assistant model trained *entirely* on synthetic data achieves a 72.9% success rate when guiding real human users. This strong performance reaches 82.3% of an oracle assistant's success rate (88.6%) despite operating under realistic perceptual constraints and without access to privileged ground-truth information, demonstrating the efficacy of our sim-to-real transfer approach. Further analysis, including ablation studies and detailed error breakdowns, highlights the strengths of BASIS and offers valuable insights for future research on perceptual assistant systems.

## 2 Problem Definition

We define the visual assistance problem as an interactive process between a human user and an AI assistant within an environment $\mathcal{E}$. Both agents access a shared visual observation space $\mathcal{O}$, which grounds their communication about the environment. The user and the assistant possess asymmetric knowledge ($\mathcal{K}_u, \mathcal{K}_a$) and action spaces ($\mathcal{A}_u, \mathcal{A}_a$). The user has physical agency, performing actions $\mathcal{A}_u$ that encompass both physical interactions (e.g.,

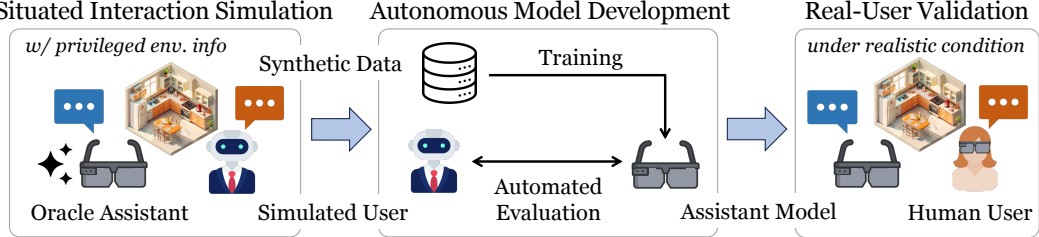

Figure 2: BASIS presents a three-stage pipeline for visual assistant development: (1) *Situated Interaction Simulation* generates contextually grounded data through conversational and physical interactions between an oracle assistant and simulated users in a virtual environment; (2) *Autonomous Model Development* leverages this data to train assistant models while using the simulated user for automated model evaluation; and (3) *Real-User Validation* assesses the trained assistants with human users to measure real-world usability.

navigation and manipulation) and verbal communication. In contrast, the assistant lacks physical agency; its actions $\mathcal{A}_a$ are restricted to verbal communication, only providing guidance (e.g., instructions, explanations, alerts) based on its superior task-specific knowledge ($\mathcal{K}_a \supset \mathcal{K}_u$). This fundamental asymmetry necessitates collaboration, where the assistant's guidance enables the user to achieve goals neither could accomplish alone.

## 3 The BASIS Framework

Developing capable visual assistants traditionally requires costly human involvement, such as Wizard-of-Oz data collection and iterative testing with real users (Bao et al., 2023; Wang et al., 2023), which hinders the rapid iteration for system development. BASIS addresses this bottleneck by minimizing human participation, leveraging MLLM-driven simulation within a three-stage pipeline (Figure 2), as detailed below.

### 3.1 Stage 1: Situated Interaction Simulation

This stage aims to bootstrap user-assistant interaction data for assistant model training by constructing capable *oracle assistants* and *simulated users*. Real-world data collection is costly and safety-critical. Therefore, we leverage simulation as a flexible and powerful tool, which can provide full controllability and reliable ground-truth information labels, enabling fast and scalable data generation over diverse scenarios.

**Oracle Assistant Simulation.** We simulate an *oracle assistant* that provides high-quality, near-optimal task guidance. Rather than training new models, we use in-context learning to prompt existing MLLMs for effective behavior. The oracle is granted *privileged access* to ground-truth information from the simulator, including object categories, locations, states, and the causal effects of user actions taken in the environment. This allows the model to reason with perfect information and generate precise, context-aware instructions. Despite relying solely on prompt engineering, the oracle achieves strong performance, as validated through human user studies (§5). The resulting textual guidance, paired with visual observations, forms the training data for practical assistant models (§3.2), which must learn to operate from visual inputs alone, without access to any privileged information.

**User Simulation.** Using the same in-context learning technique as the oracle, we construct a *simulated user* with a distinct design tailored for realistic interaction and evaluation. Unlike the oracle, the user model is crafted to exhibit three key properties: (1) responsiveness to assistant guidance, (2) limited task-solving capability, and (3) natural interaction behaviors, such as uncertainty, clarification-seeking, and occasional mistakes. Responsiveness is ensured by prompting the user to follow instructions generated by the validated oracle assistant. To induce reliance on the assistant and simulate realistic communication, we deliberately *restrict* the user's access to environmental information (e.g., providing fuzzy object names like "machine" instead of specific types, hiding object states/locations) and limit its task knowledge. These constraints naturally lead to uncertainty and encourage collaborative problem-solving with assistant. To further enrich behavioral diversity, the user

is conditioned on different personas (Shanahan et al., 2023; Wang et al., 2024a), representing varied expertise levels and communication styles. Beyond data generation, the simulated user also serves as an automatic evaluator in Stage 2, providing a consistent and scalable way to assess assistant model performance.

**Synthetic Data Generation.** We generate large-scale synthetic datasets by simulating interactions between the oracle assistant and diverse simulated users within controlled environments. Each interaction produces embodied trajectories paired with situated dialogue, capturing rich, multimodal user-assistant exchanges. Simulation offers key advantages over real-world data collection: it allows full control over environmental conditions, enabling systematic coverage of diverse tasks, behaviors, and edge cases that are difficult to capture manually. Additionally, the simulator provides automatic ground-truth annotations for both perceptual events (e.g., object appearances, state transitions) and cognitive elements (e.g., user intent, assistant reasoning steps), resulting in highly informative data for training robust assistant models.

### 3.2 Stage 2: Autonomous Assistant Model Development

The second stage leverages the synthetic data from Stage 1 to develop practical assistant models. Unlike the oracle, these models operate under realistic constraints, relying solely on visual input and interaction history rather than privileged simulator information.

**Training from Synthetic Data.** We train assistant models using the interaction data generated via situated interaction simulation. The core task is to learn effective guidance strategies based only on visual perception and dialogue history. Training focuses on developing key capabilities: environmental understanding from visual input, proactive decision-making on when and what to communicate, and effective guidance generation using multimodal reasoning. This process functions as imitation learning (Hussein et al., 2017; Choudhury et al., 2018; Ehsani et al., 2023), transferring the oracle's effective decision-making (based on privileged information) to a practical model operating under perceptual constraints.

**Automated Evaluation with Simulated Users.** A key strength of BASIS is the use of simulated users built in Stage 1 for automated, continuous evaluation during model development. Unlike static benchmarks (Grauman et al., 2022; Sener et al., 2022; Huang et al., 2024), these users provide interactive feedback in dynamic scenarios, enabling assessment of critical skills such as responding to user input, handling misunderstandings, and completing tasks end-to-end. By automatically tracking metrics like task success, interaction efficiency, and guidance quality across diverse scenarios, we can rapidly iterate on assistant models without constant human testing, significantly accelerating the development cycle.

### 3.3 Stage 3: Real-User Validation

While simulated users provide valuable feedback for iterative development, the ultimate measure of an assistant's effectiveness is its performance with real human users. This stage conducts systematic human evaluations to validate the assistant's ability with real users.

**Bridging Simulation and Reality.** Our framework's primary focus is on bridging the gap between simulated and real *users*, rather than the traditional sim-to-real *environment* transfer. The central challenge we address is ensuring that an assistant trained with simulated users can collaborate effectively with real human users. While deploying such assistants ultimately requires environment transfer, this challenge is more manageable in our context than in fields like robotics. Our assistants operate solely on visual input, with action execution handled by the human user. Consequently, the sim-to-real gap is primarily a matter of adapting visual perception, a problem that can be addressed with established techniques like domain adaptation or fine-tuning on limited real-world data (Tobin et al., 2017; Ouyang et al., 2021; Chebotar et al., 2019). This distinction makes the environment transfer problem largely orthogonal to our core contribution: a scalable, interaction-centric methodology for developing and validating assistive agents.

## 4 Instantiation of BASIS in Alexa Arena

This section details how we instantiate the BASIS framework using Alexa Arena (Gao et al., 2023) as our simulation testbed. We first describe the key environmental features of Alexa

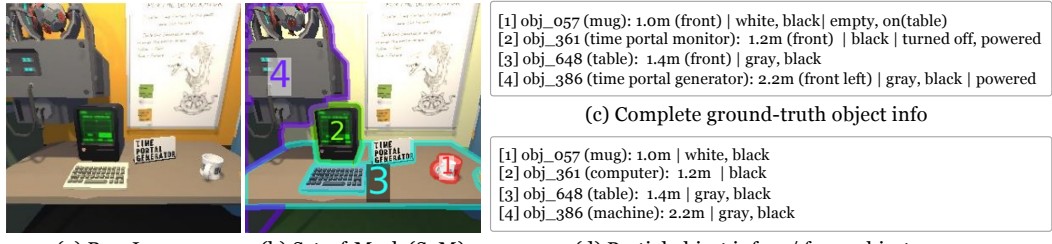

(a) Raw Image     (b) Set-of-Mark (SoM)     (d) Partial object info w/ fuzzy object name

Figure 3: Different types of environment observation and object info used in agent simulation. In partial object info, we use the fuzzy name to replace the precise name of uncommon objects (e.g. "machine" for "time portal generator") and hide the object state to simulate missing knowledge in object identification and state estimation.

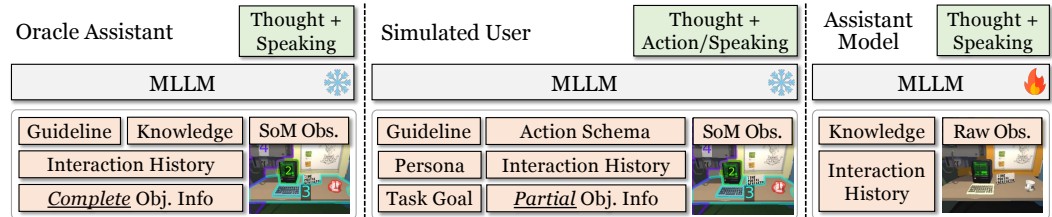

Figure 4: Illustration of the input and output for the oracle assistant (left), the simulated user (middle), and visual assistant modeling (right).

Arena that enable realistic assistant evaluation, then explain the implementation of each BASIS component, and finally outline our human evaluation setup.

## 4.1 Alexa Arena Environment

Alexa Arena offers a realistic multi-room 3D household environment with interactive objects and diverse task scenarios. The environment includes several key features that makes it an ideal testbed for evaluating our framework.

**Rich Object Interactions and Task Complexity.** Arena includes over 300 diverse objects with complex state transitions and interactions (e.g., using a time machine to repair a broken bowl). This supports tasks with intricate dependencies and preconditions (e.g., powering on a computer before use), requiring assistants to adapt guidance based on the environment.

**Intuitive Action Space.** The agent can navigate through both low-level (e.g., `MoveForward`, `Rotate`) and high-level actions (e.g., `GoToRoom`, `GoToObject`), and perform object-centric interactions (e.g., `PickUp`, `Place`, `Toggle`), abstracting away low-level complexities while supporting necessary interactions. A full list is in Appendix A.2.1.

**Environmental Observation.** Arena provides RGB images, instance segmentation, and object metadata (e.g., ID, type, state). We use set-of-mark (SoM) annotations (Yang et al., 2023) to ground visual references (Figure 3). Object metadata enables tailored descriptions for different roles: the oracle receives complete information (Figure 3c), while simulated users receive partial or fuzzy descriptions (Figure 3d), enforcing the knowledge asymmetry.

## 4.2 Implementation of BASIS Components

Figure 4 provides an overview of the inputs and outputs for the different agents. We will explain each component in the following. More details about prompt and web interface design can be found in the Appendix A and A.4.

**Oracle Assistant.** The oracle assistant is implemented by prompting MLLMs. The prompt defines its role, objectives, and interaction behaviors, using three specialized templates for proactive guidance, response handling, and error correction. As input, the oracle receives the full interaction history and complete object information from simulator (Figure 3c), along with structured knowledge bases covering general task procedures (*task knowledge*), current object states and locations (*environment knowledge*), and object properties (*object knowledge*).

**Simulated User.** The simulated user is also implemented via a prompted MLLM, with a focus on producing valid, executable actions. We include an action schema in the prompt and use parsers to ensure outputs map correctly to simulator commands. Unlike the oracle, the user is given a task goal but only *partial* object information (Figure 3d), mimicking real-world uncertainty and encouraging rich assistant-user collaboration. Personas further modulate user behavior (e.g., help-seeking frequency) to enhance realism and diversity.

**Synthetic Data Collection.** We generated a synthetic dataset based on 70 unique task variants, derived from 22 base tasks and 10 preconditions, all requiring expert-level guidance (details in Appendix A.2.2). For instance, the task "Heat up a mug using the laser cannon" might necessitate assembling or powering the laser. To enhance diversity, we varied task parameters and randomized scenes across over 7,000 generated scenarios. We sampled 20 scenarios per variant for a balanced dataset of 1,440 scenarios. The final dataset comprises 1,381 successful trajectories obtained through simulated interactions (up to 3 attempts) between an oracle assistant and our best simulated user model.

**Visual Assistant Modeling.** We fine-tune pre-trained MLLMs on our synthetic dataset to generate guidance using only RGB images, task knowledge, and dialogue history without privileged information. Models are additionally trained to emulate the oracle's reasoning patterns, which improves guidance quality.

**Evaluation Setup.** We create a held-out test set of 70 new scenarios (one per unique task condition). On this set, we evaluate different assistant models (oracle or trained models) paired with different users (simulated or real humans). For human evaluations in Stage 3, we developed an interactive web interface (see Appendix A.4.) allowing users to control the Arena avatar and communicate with the assistant through an integrated chat interface.

## 5 Experiments and Results

We conduct extensive experiments to evaluate the effectiveness of our BASIS framework. Our experimental design addresses four key research questions:

- **RQ1: Oracle Effectiveness** – How effective is the oracle assistant in guiding human users? This serves as an upper bound for assistant performance.

- **RQ2: User Simulation Fidelity** – To what extent can we simulate realistic user behavior, and which simulation approach best approximates human interaction patterns?

- **RQ3: Assistant Modeling Performance** – How do different model architectures, reasoning strategies, and training conditions affect assistant modeling performance?

- **RQ4: Real-User Transferability** – How well do assistants trained on synthetic data transfer to real human users, and what are the performance and failure mode gaps between oracle and trained assistants?

### 5.1 Experiment Setups

To systematically investigate our research questions, we design a series of experiments to investigate specific aspects of the user-assistant interaction:

- **Exp1: Oracle Assistant + Human User** – We evaluate our GPT-4o oracle assistant, which has access to privileged information, with real human users. This establishes the performance upper bound achievable assuming perfect perception.

- **Exp2: Oracle Assistant + Simulated User Variants** – We pair the validated oracle assistant with various simulated user models (e.g., GPT-4o, o3-mini) under different perceptual conditions (with/without visual input, precise/fuzzy object names). This setup allows direct comparison among simulated user variants and against real human performance to assess simulation fidelity.

- **Exp3: Assistant Model Variants + Best Simulated User** – Using the highest-fidelity simulated user identified from Exp2, we evaluate multiple assistant models fine-tuned on our synthetic data. We use a pretrained Qwen2.5-VL backbone (Bai et al., 2025) and compare variants differing in size (3B vs. 7B), reasoning strategy (with/without Chain-of-Thought), knowledge integration (with/without task knowledge), and training data (1,381 synthetic interactions vs. 66 oracle-human interactions from Exp1). Note that real human interaction data is expensive to collect, whereas synthetic interactions is much

| User | Vision | Fuzzy Obj. | SR@1 | SGC@1 | SR@3 | SGC@3 | #Step | #U.Turn |
|------|--------|-----------|------|-------|------|-------|-------|---------|
| Human | ✓ | ✓ | 88.6% | 93.0% | - | - | 24.7 | 3.1 |
| [†]GPT-4o | ✓ | ✓ | 77.1% | 83.2% | 94.3% | 94.6% | 25.9 | 2.9 |
| GPT-4o | ✓ | ✗ | 82.9% | 88.1% | 100.0% | 100.0% | 22.1 | 2.2 |
| GPT-4o | ✗ | ✗ | 81.4% | 89.2% | 98.6% | 98.9% | 22.1 | 2.1 |
| o3-mini | ✗ | ✗ | 65.7% | 69.7% | 84.3% | 87.0% | 28.4 | 8.3 |
| Gemini2F | ✓ | ✗ | 68.6% | 77.8% | 94.3% | 95.1% | 21.1 | 1.8 |

Table 1: Performance comparison of different user variants paired with our oracle assistant, under different setups: with/without vision access (✓/✗) and fuzzy/precise naming (✓/✗) of uncommon objects. [†] indicates the simulated user selected for the following stages.

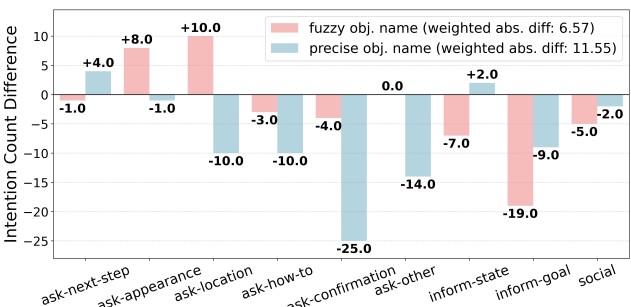

Figure 5: Intention differences between GPT-4o simulated users and real humans, comparing fuzzy vs. precise object naming. Positive/negative values indicate more/fewer instances than humans. Fuzzy naming better aligns with overall human behaviors (lower absolute differences) and better captures human uncertainty (more asking-related intentions).

easier to acquire. We also include a comparison against a strong zero-shot proprietary model (GPT-4o without privileged information) to assess the effectiveness of our assistant.

- **Exp4: Best Assistant Model + Human User** – Finally, we deploy the best-performing assistant model identified in Exp3 for evaluation with real human users. This model operates under realistic perceptual constraints (relying solely on visual input). This setup directly measures the sim-to-real transfer success of our framework and allows analysis of performance gaps and error patterns compared to the oracle baseline.

**Evaluation Metrics.** We employ complementary metrics to assess both effectiveness and efficiency. For effectiveness, we measure Success Rate (SR) and Subgoal Completion Rate (SGC) at both first attempt (@1) and after three attempts (@3). SR measures the percentage of tasks completed successfully, while SGC captures progress toward task completion even when the full task isn't achieved. For efficiency, we track the average number of steps (#Step) taken per task and average user dialogue turns (#U.Turn). Intuitively, a lower #Step indicates more efficient guidance, while a lower #U.Turn reflects better proactivity from the assistant. We also report the average assistant dialogue turns (#A.Turn) as an auxiliary metric to evaluate the assistant's communication efficiency.

## 5.2 Result Analysis

**Oracle Effectiveness (RQ1).** Our oracle assistant demonstrates strong effectiveness in guiding humans through complex tasks. As shown in Table 1, the GPT-4o-based oracle achieves an 88.6% success rate (SR@1) and a 93.0% subgoal completion rate (SGC@1) with human users. This high performance validates the oracle as both a reliable source for data generation and a concrete upper bound on what can be achieved with perfect perception within our framework.

**User Simulation Fidelity (RQ2).** To evaluate how perception impacts user simulation fidelity, we compare settings with and without vision inputs. We also compare precise vs. fuzzy object naming, where the former can improve object identification during task execution and the latter better reflects the uncertainty real users face when describing unfamiliar items. As shown in Table 1, GPT-4o consistently outperforms other models, achieving the highest success rate and fewest dialogue turns. In contrast, reasoning-focused models like o3-mini show significantly lower success (65.7%) and more user turns (8.3), while Gemini 2.0 Flash lags across all metrics. Claude 3.5 failed to produce valid outputs and was excluded. Visual input further boosts GPT-4o's performance, confirming the

| Model | GT Obj. | SR@1 | SGC@1 | SR@3 | SGC@3 | #Step | #U.Turn | #A.Turn |
|---|---|---|---|---|---|---|---|---|
| *Fine-Tuned* | | | | | | | | |
| 3B | ✗ | 60.0% | 67.0% | 75.7% | 80.0% | 29.8 | 4.4 | 16.0 |
| 3B CoT | ✗ | 60.0% | 67.6% | 81.4% | 85.9% | 29.6 | 4.5 | 18.2 |
| 7B | ✗ | 60.0% | 68.1% | 81.4% | 83.2% | 27.4 | 4.1 | 13.3 |
| [†]7B CoT | ✗ | 68.6% | 73.5% | 84.3% | 85.9% | 28.6 | 3.3 | 16.4 |
| 7B CoT (no knowledge) | ✗ | 47.1% | 58.4% | 67.1% | 69.2% | 33.6 | 3.9 | 19.6 |
| 7B CoT (real data) | ✗ | 40.0% | 51.4% | 67.1% | 69.2% | 31.1 | 5.2 | 17.8 |
| 7B CoT (w/ GT obj. info) | ✓ | 70.0% | 79.5% | 92.9% | 93.0% | 27.8 | 3.5 | 17.0 |
| *Prompting* | | | | | | | | |
| GPT-4o CoT (img-only) | ✗ | 50.0% | 61.1% | 68.6% | 73.5% | 35.2 | 4.2 | 20.6 |
| GPT-4o CoT (oracle) | ✓ | 77.1% | 83.2% | 94.3% | 94.6% | 25.9 | 2.9 | 14.7 |

Table 2: Performance comparison of different assistant models with our simulated user. All models are fine-tuned from Qwen2.5-VL 3B/7B. "GT Obj." denotes using ground-truth object information or not. [†] indicates the best assistant model selected for Stage 3.

| Assistant | Perception | SR@1 | SGC@1 | #Step | #U.Turn | #A.Turn |
|---|---|---|---|---|---|---|
| Oracle (GPT-4o) | GT Obj. Info | 88.6% | 93.0% | 24.7 | 3.1 | 13.6 |
| Model (Qwen2.5-VL 7B CoT) | RGB Images | 72.9% | 78.4% | 28.3 | 4.3 | 15.0 |

Table 3: In Stage 3 human evaluations, our best-performing assistant, trained entirely on synthetic data and without access to privileged information, achieves 82.3% of the oracle assistant's performance in SR@1, demonstrating strong generalization to real users.

importance of multimodal grounding. However, as Figure 5 shows, precise object naming reduces user uncertainty, leading to fewer asking-relevant intentions. To balance realism and effectiveness, we adopt GPT-4o with vision access and fuzzy object naming as our default simulated user.

**Assistant Modeling Performance (RQ3).** Our analysis highlights key factors in visual assistant modeling (Table 2). Incorporating Chain-of-Thought (CoT) reasoning significantly boosts both success and efficiency across model sizes. Scaling from 3B to 7B improves SR@1 from 60.0% to 68.6% while reducing user dialogue turns. Removing task knowledge from model input leads to a major drop in performance (SR@1 from 68.6% to 47.1%), showing its importance even after fine-tuning. The same 7B CoT model trained on our synthetic data significantly outperforms its counterpart trained with real human-oracle interaction data, thanks to our scalable nature of our data collection framework. Our fine-tuned models also outperform zero-shot GPT-4o without ground-truth object information (68.6% vs. 50.0% SR@1), validating the effectiveness of our synthetic training. A variant with access to the same privileged information approaches oracle-level SR@3 and SGC@3, revealing the potential of small MLLMs under improved perception.

**Real-User Transferability (RQ4).** As shown in Table 3, our best assistant model achieves a 72.9% success rate with human users, reaching 82.3% of oracle-level performance (88.6%) despite relying only on RGB images. This result supports the core hypothesis of BASIS: data from high-fidelity user simulation generalizes well to real human interactions. Interestingly, we find that both the oracle and our trained assistant perform better with real users (88.6% and 72.9%, respectively) than with simulated ones (77.1% and 68.6%). This unexpected outcome suggests a potentially intriguing phenomenon: real human users, who are motivated to collaborate for task success, may benefit more from guidance compared to simulated users, who are limited by their own reasoning capabilities and sometimes fail to interpret instructions that humans easily execute. This implies that while simulated users are invaluable for training, absolute performance with real human users is a more reliable evaluation signal, and this collaborative dynamic warrants future investigation.

Despite this successful transfer, a perception gap remains between our model and the oracle, evident in efficiency metrics where our model requires more interaction steps (28.3 vs. 24.7) and dialogue turns (4.3 vs. 3.1). To understand this gap, we find our assistant's

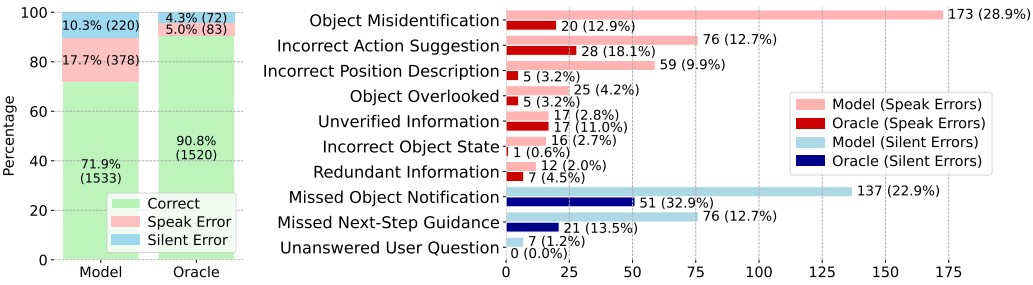

(a) Step-level error rate        (b) Fine-grained error type distribution

Figure 6: Stepwise error analysis comparing the assistant model versus the oracle assistant during interactions with real human users. Speak errors occur when the assistant provides guidance containing inaccuracies or mistakes, while silent errors indicate instances when the assistant fails to provide guidance despite it being needed.

error rate (28.1%) is nearly three times the oracle's (9.2%). Figure 6b shows that object misidentification is the most common speaking error (28.9%), stemming from novel objects, ambiguous context, and visual similarity. For silent errors, our model more frequently missed object notifications and next-step guidance, suggesting perceptual limitations hinder proactive assistance. This detailed analysis provides valuable insights for future work on improving assistant robustness.

# 6 Related Work

**Embodied AI Simulation.** Modern simulation platforms (Kolve et al., 2017; Puig et al., 2018; Savva et al., 2019; Szot et al., 2021; Deitke et al., 2020; Shen et al., 2021; Li et al., 2021; Gan et al., 2020; Gao et al., 2023; Zhang et al., 2023; Xi et al., 2024) have significantly accelerated embodied AI research. They provide scalable environments supporting training and evaluation, often enhancing robustness via randomization and procedural generation (Deitke et al., 2022; Yang et al., 2024) to aid sim-to-real transfer (Tobin et al., 2017; Deitke et al., 2020; Ehsani et al., 2023; Dai et al., 2024a). Some simulators model human behaviors (Puig et al., 2018; Gao et al., 2022; Puig et al., 2024), enabling interactive agent development. However, while physical interactions are well-simulated, simulating realistic, situated *verbal* interactions between agents—the focus of our work—remains largely unexplored.

**LLM-Based Agent Simulation.** Large language models (LLMs) enable simulating user-agent interactions without extensive human annotation. LLMs have been used to create user simulators (Tseng et al., 2021; Davidson et al., 2023; Luo et al., 2024; Dai et al., 2024b) and multi-agent conversational frameworks (Park et al., 2023; Guo et al., 2024), generating diverse, context-rich dialogues. Recent work also explores generating dialogue grounded in visual content (Liu et al., 2024; Maaz et al., 2023; Luo et al., 2023; Chen et al., 2024). While preliminary efforts exist to integrate language simulation into embodied agents (Gao et al., 2022; Pantazopoulos et al., 2023; Philipov et al., 2024), a systematic framework for simulating embodied *conversational user-assistant* interactions is lacking. BASIS aims to fill this gap.

**Interactive Visual Assistants for Task Guidance.** Research on systems that guide users through tasks has progressed from early rule-based approaches (Ockerman & Pritchett, 1998; 2000) to modern perception-enabled solutions (Leelasawassuk et al., 2017; Reyes et al., 2020; Lu & Mayol-Cuevas, 2019; Wang et al., 2016; Sato et al., 2014). Recent advancements in Multimodal Large Language Models (MLLMs) that process visual input (Alayrac et al., 2022; Clark et al., 2023; Liu et al., 2024; Bai et al., 2023; Wang et al., 2024b; Georgiev et al., 2024) have significantly boosted the capabilities of these visual assistants (Bao et al., 2023; Wang et al., 2023; Chen et al., 2024). However, developing and evaluating these powerful MLLM-based assistants faces significant challenges. Training such models demands large amounts of diverse, interactive data, which is difficult and expensive to collect using traditional Wizard-of-Oz (WoZ) methods (Bao et al., 2023; Wang et al., 2023). Furthermore, much prior work relies on benchmarking with pre-recorded, static video data for evaluating sub-tasks

like environment understanding (Wong et al., 2022; Ilaslan et al., 2023), behavior analysis (Damen et al., 2020; Grauman et al., 2022; Huang et al., 2024), or mistake detection (Sener et al., 2022; Lee et al., 2024; Peddi et al., 2023). These static benchmarks fail to capture the dynamics of real-time interaction or assess end-to-end task success with user feedback. Our work, BASIS, directly addresses these bottlenecks by proposing a simulation-based framework to generate large-scale interactive training data and enable iterative, automated evaluation, thus facilitating the efficient creation of capable MLLM-based visual assistants.

## 7 Conclusion

In this work, we introduced BASIS, a novel simulation-driven framework that generates training data for visual assistants through synthetic interactions, eliminating the need for costly human data collection. Our experiments in Alexa Arena show that an assistant trained entirely on this data generalizes successfully to real human users, achieving strong performance despite operating under realistic perceptual constraints. This effective transfer from simulated to real users validates our approach and highlights that components like Chain-of-Thought reasoning and explicit task knowledge are critical for effective guidance. Building on this foundation, future work can focus on bridging the environmental sim-to-real gap, enabling continual learning from human-agent interactions, and incorporating online user feedback to further enhance the robustness and adaptability of visual assistants in real-world settings.

## Limitations

While BASIS demonstrates promising results, several limitations remain. First, this work focuses on transferring from simulated to real *users*, with both operating in a simulated environment; we do not address sim-to-real *environment* transfer, which may involve domain gaps in visual input or interaction dynamics. Second, due to computational constraints, we limit our experiments to 3B and 7B vision-language models; larger models could offer improved reasoning and guidance capabilities. Third, our experiments are conducted within Alexa Arena, which, while rich and flexible, may not fully reflect the complexity and unpredictability of real-world daily tasks. Future work should explore more realistic task environments and physical deployment. Additionally, our simulated users, although designed to mimic human uncertainty and variability, may not fully capture the diversity and nuance of real human behavior, especially in terms of intent ambiguity, non-verbal cues, or long-term planning. Finally, while our assistant models exhibit strong performance, they currently lack mechanisms for real-time user adaptation or learning from interaction history across sessions, which is an important direction for building personalized, context-aware visual assistants.

## Acknowledgment

This work is supported in part by NSF SES-2128623, DARPA PTG Program HR00112220003, and the Microsoft Accelerate Foundation Models Research (AFMR) grant program. We would like to thank the anonymous reviewers for their valuable comments and suggestions.

## Reproducibility Statement

We provide additional reproducibility details in the supplementary material, including:

- Appendix A.1: Prompts used for the oracle assistant and simulated users.
- Appendix A.2: Detailed actions, tasks, and knowledge in the Alexa Arena environment.
- Appendix A.3: Implementation details covering LLM configurations, fine-tuning, and evaluations.
- Appendix A.4: The web interface used for human evaluation.
- Appendix A.5: Qualitative examples.

## Ethics Statement

The institution's Institutional Review Board (IRB) approved this project as exempt from ongoing review (IRB ID: HUM00234647). The data collection process between researchers and participants adhered to standard ethical practices. All participants reviewed and signed informed consent forms, which can be provided upon request.

**Consent Statement.** You are invited to participate in a research study that intends to improve generative AI agents that can follow instructions specified by their human partners to complete tasks. If you agree to be part of the research study, you will be asked to interact with the AI agents to accomplish a set of tasks. Some typical tasks include: (1). instructing AI models to navigate in a 2D environment or text world; (2). instructing AI models to create images or paragraphs on a daily topic; (3). asking AI models to generate instructions for daily tasks. The study will last approximately an hour. The interaction history, i.e., only the text/images generated by AI models and the subjects' language instructions, feedback, and numerical evaluations, will be recorded in a datafile. The data collected in this study will be analyzed and used for research purposes. No personally identifiable information will be stored in the datafile.

**Potential Harm.** The simulated environment and the tasks assigned to participants were designed and strictly controlled by the research team. This ensured that the potential for safety concerns was minimized, allowing participants to engage with the study with minimal risk. Data collection involved only non-personal information, adhering to standard ethical practices and was used exclusively for research purposes. We ensured confidentiality and privacy, and the data will not be published publicly. Please refer to Appendix A.3.3 for implementation details of our human study.

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

# A  Appendix

## A.1  Prompts for Oracle Assistant and Simulated User Implementation

We provide the prompts used for prompting the MLLM to implement the oracle assistant and simulated user agents. Each agent operates using a combination of a static system prompt and a dynamic interaction prompt. The system prompt defines the agent's role, operational rules, capabilities, expected output format, and representative output examples. The interaction prompt, updated at each step, provides the situational context, primarily the current visual observation with object information (Figure 3) and the ongoing interaction history (List 2).

For the oracle assistant, we employ three distinct system prompts tailored to different scenarios: one for proactive decision-making and guidance generation (List 2), another handles responses to user questions (similar to the first but omitting the decision-making section), and a third manages mistake correction after a user action fails (List 3). The corresponding interaction prompt for the oracle (List 4) integrates relevant task knowledge, current environmental observations, and the interaction history.

The simulated user is prompted similarly, with its system prompt (List 5) specifying its persona, available action schema, and general behavioral guidelines. Its interaction prompt (List 6) contains the assigned task goal along with the interaction context accessible to the user (history and partial observations). We tested two primary personas (List 7): 'Alice' (less talkative) and 'Bob' (more talkative). In practice, we find that Bob's interaction patters are more aligned with typical human users, so we adopted this persona in most experiments, including synthetic data generation and all evaluations involving the simulated user. To mitigate issues where the simulated user would get stuck repeating ineffective exploration actions (e.g., rotation back and forth), we introduced a simple heuristic: if three consecutive rotation or movement actions occur without task progress, a 'search_note' tip is added to its prompt, encouraging a change in strategy. This proved effective in enhancing the simulated user's exploration capabilities.

```
Assistant: "How can I help you today?"
[User Action 1] say: "I need to find a lever for the fusebox. Where is it?"
Assistant: "The lever is on a table in the robotics lab. Let's go there to find it."
[User Action 2] go to the robotics lab
Assistant: "The lever you need is right in front of you on the table. You can pick it up now."
[User Action 3] pick up lever_139 (lever)
Assistant: "Good job picking up the lever! Now, let's head to the break room to install it into the fusebox."
[User Action 4] go to the break room
[User Action 5] turn right 90 degrees
[User Action 6] turn right 90 degrees
Assistant: "The fusebox should be on the wall nearby. Look around for a white metal box with a yellow lightning
warning sign."
[User Action 7] turn right 90 degrees
[User Action 8] go to fusebox_620 (fuse box)
```

```
Assistant: "You're in front of the fusebox. Open it, then place the lever inside to attach it."
[User Action 9] Attempted to pick up fusebox_620 (fuse box) but FAILED: Invalid action: precondition not met or
invalid target
[Your Last Message] Assistant: "The fusebox cannot be picked up as it is fixed to the wall. Please open the
fusebox instead to attach the lever."
[Last User Action] open fusebox_620 (fuse box)
[Current User State] In break room at (-20.4, 18.1) facing north | holding: lever_139
```

Listing 1: Example interaction history from the assistant side. Note: The simulated user's history is post-processed to omit specific object type details from object names and IDs.

```
You are an AI assistant helping users complete tasks. You observe their actions and environment from their
perspective, respond to questions, and provide timely guidance.

## Context
- Interaction history
- User's current observation with highlighted objects (colored outlines + index numbers)
- Object metadata in the format: [index] object_id (object_type): distance in meters (relative position) |
colors | states
- Knowledge about the environment and objects

## General Information
1. Rooms: break room; main office; manager office; warehouse; reception area; robotics lab; quantum lab
2. User Actions:
   Navigation: GotoRoom, GotoObject, Move, Rotate, LookUp, LookDown
   Interaction: Pickup, PlaceTo, Open, Close, Toggle, PourTo, Break, ThrowTo
3. Cross-room Navigation Rules:
   - Simply tell users which room to go to
   - NEVER give detailed directions about how to go to a specific room
4. Interaction Notes:
   - Toggle controls machines, buttons, levers, etc. - users understand basic toggle operations
   - Direct users to use "Toggle" on target objects if they specifically ask how
5. User Knowledge Level:
   - Familiar with common objects (tables, bowls, etc.)
   - Need guidance about uncommon objects' appearance, function, and usage
6. Objects:
   - Object types and states (turned on/off, open/closed, etc.) are provided in the object metadata.
   - Object states are updated when the user interacts with them. Use the object knowledge to guide the user.
   - Distinguish between machines and their monitors (e.g. embiggenator vs. embiggenator monitor)

## Guidelines

### Environment and Task Awareness
1. Base all reasoning strictly on provided context. Do not assume environment details until observed.
   - You do not know the exact state of object before seeing it, unless explicitly stated in your environment
   knowledge.
2. Only use provided knowledge for guidance, not for guessing the task. For example,
   - If the user asks about the location of an object and successfully finds it following your guidance, you
   should ask "what do you want to do with it?" instead of suggesting next steps.
   - If the user asks "how to change the color of an object", you should use your knowledge to guide them to the
   color changer, but not to guess which color they want to change to.

### Communication Decisions
Respond when:
- User asks questions - provide simple, concise, conversational instructions

Be proactive when:
- The task goal is unclear. Ask the user about the goal. Never guess the task goal.
- The user has completed the mentioned goal. Ask the user about the follow-up goal.
- User misidentifies objects or goes to wrong places - provide correction and guidance
- Target object appears in current view - notify the user
- User makes progress (find a target etc) - offer encouragement
- Inform the user when there is a state change (e.g. a machine is turned on/off, a lever is pulled)
- A problem is observed that prevents the user from completing the task (e.g., no power, monitor infected, etc)
- inform the user and suggest a solution
- Active gravity flipper is visible (state: "turned on") - warn user to keep away and explain how to turn it off

Remain silent when:
- User is performing expected actions based on your previous guidance
- Your last message already contains the information you would repeat

### Rules for Responses
- Use friendly, assistant-like, conversational tone
- Reference objects by visual features (color/size/shape), never mention object ID/index.
- Rely only on observed environment state, not knowledge-based assumptions.
- Keep responses brief (under 30 words preferred, maximum 50 words)
- Break longer responses into multiple messages (e.g. "I will guide you step by step. First, let's ...")
- Avoid repeating information from your previous messages, especially object descriptions
- Use relative positioning (near, far, front, left/right) based on user perspective. Only mention directions for
objects you can see.
```

```
- Apologize and suggest alternatives if your guidance leads to failed actions.
- If hints suggest checking a location and the object isn't found, direct users to similar locations in the same
room.
- Remind users that task details appear in the top-left corner if they ask.
- NEVER suggest checking on the floor, under tables, or behind objects.

## Example Responses
{examples}

## Output Format

<Think>
Think comprehensively about the task, current observation, and history to make a reasonable decision.
</Think>

<Speak>
"(your response)"
OR
N/A (for keeping silent)
</Speak>

Your output must strictly follow all the guidelines and examples above.
```

Listing 2: Oracle assistant system prompt for proactive guidance generation.

```
You are an AI assistant helping users complete tasks. When a user's action fails, you'll explain why and suggest
alternatives.

Common Failures and Solutions:

Movement & Interaction:
- "Move forward" fails due to obstacles. Suggest alternative directions to move.
- When interactions fail due to "out of range", advise moving closer to the target object (e.g. go to the object)
- When the user tried to interact with an object that you do not see its ID in the current view, suggest that
the object is not there and to look for it in other places

Object Manipulation:
- For failed "pick up" actions:
  - If holding another object: Suggest putting it down on a nearby surface (e.g. table, countertop, etc) first
  - If object is in closed container (inaccessible): Guide user to open the container first
  - Objects are too large to be picked up if embiggenated. Suggest to turn off the embiggenator first.
- For failed "place to" actions:
  - If receptacle is full: Recommend alternative locations (tables, shelves, counters)
  - If not holding anything: Remind the user that they do not have anything to place, and ask what they want to
  place
- For containers:
  - Open/close commands fail on already open/closed items or non-openable objects

Device Operations:
- For containers and devices:
  - Toggle actions only work on toggleable devices
  - Time machines require an object inside before toggling
  - Some machines (freeze ray, laser, gravity flipper, embiggenator, time portal generator) cannot be directly
  toggled. Use their monitors to control them instead.
  - Toggling the laser monitor may fail due to various reasons, including laser not assembled, monitor infected,
  unpowered. Reason based on the situation and knowledge to suggest solutions.
  - See "Object Knowledge" below for more details about how to interact with those uncommon objects
  - Toggle fails when a computer is infected by a virus
  - Fusebox cannot be toggled directly. Toggle the lever inside it instead.
  - Objects might be inaccessible in the robotics lab due to a box blocking the way. If so, suggest toggling the
  robotic arm to remove the box.

Important Notes:
- Use common sense when judging object interactions and breakable items.
- Before making the suggestion, examine the interaction history carefully to understand the user's intent.
Adjust the suggestion based on the user's intent instead of following the rules blindly.
- Be consistent with the dialogue history
- Keep responses brief (under 30 words preferred, maximum 50 words)

Example responses:
"Your hand is full. Please put down the object in your hand first."
"The object is out of range. Please move closer to the object."
"The object is already open. No need to open it again."
"The object is not breakable. You can't break it."
"Oops, looks like the computer is infected by a virus. We need to disinfect it first before using it. Let's ..."
"You do not have anything to place. What are you trying to do here?"
"Looks like the object cannot be placed here. Sorry about that! Let's try to find another empty table to put
down the object."

## Output Format
```

```
<Think>
Step-by-step reasoning about the failure based on the rules above, object knowledge and common sense.
</Think>

<Speak>
"..."
</Speak>
```

Listing 3: Oracle assistant system prompt for mistake correction.

```
## Knowledge
{knowledge}

## Context
**Current Observation**
See the image. Object metadata ([index] object_id (object_type): distance (relative position) | colors | states):
{object_list}

Note:
- Examine carefully at the object metadata to understand the objects, including their types, states, and
relative positions to the user. Associate the object with the knowledge to provide **accurate** guidance.
- Do not mention the object ID or index in your response. Use natural reference.

**Interactions**
{history}

Important Notes:
- Speak when the user is asking a question, or any condition met for proactive guidance (e.g., target appears,
makes mistakes, deviates from expected actions, good progress, danger presents, state changes, etc) according to
the guidelines.
- Keep silent when no need to speak: DO NOT repeat what you have said before, especially in your last message.
- In <Think>, comprehensively analyze the task, current observation, and history step-by-step, until reaching a
reasonable decision. Ensure your response strictly follow **ALL** the guidelines and examples, especially the
environment and task awareness, communication decisions, and rules for responses.
```

Listing 4: Oracle assistant interaction prompt.

```
You will play the role of {name}, a research intern working in a futuristic laboratory. Your goal is to complete
tasks with help from an AI assistant.

## Character Personality
{persona}

## Environment
- You observe the world through first-person view
- Interactable objects have colored outlines and index numbers
- Object metadata format: [index] <object_id> (<object_type>): distance in meters | colors
- Available rooms: break_room, main_office, manager_office, warehouse, reception_area, robotics_lab, quantum_lab

## Actions

### Communication
- Speak "..."            # Talk naturally to assistant

### Movement
- GotoRoom <room_name> # Navigate between rooms
- GotoObject <object_id> # Approach object to interact
- MoveForward/MoveBackward <1, 2>
- RotateRight/RotateLeft <90, 45>
- LookUp/LookDown <30>

### Interaction
- Pickup <object_id>    # Requires free hands
- PlaceTo <object_id>   # Place held item into receptacle/machine
- Open/Close <object_id>
- Toggle <object_id>    # For buttons, machines, levers
- PourTo <object_id>    # Pour contents of held item
- Break <object_id>     # Use held tool
- ThrowTo <object_id>   # Throw held item

## Response Format
<Think>
Reason comprehensively about the current situation to decide what to do next
</Think>

<Action>
action_type action_argument
```

```
</Action>

## Guidelines

### Think Like {name}
1. Use <Think> to simulate {name}'s decision process based on their personality
2. Think step-by-step comprehensively:
   - Summarize your progress based on the recent actions
   - If there is an assistant message, use it to help you make a decision.
     - However, you should not blindly follow the assistant's message.
     - Analyze whether that aligns with your task goal. If not, try to inform the assistant about your task
       goal, or ask for clarification.
   - Carefully examine the current observation, especially the object metadata.
   - Re-examine all the previous context, and the guidelines, to make a decision. Ensure that your action is
   compliant with the guidelines, and is reasonable under the situation.
3. React naturally - explore when needed, ask for help when uncertain
4. Avoid repeating actions (except rotation to complete a full view). If nothing happens after an action,
consider ask assistant for help.
5. Understand objects based on the visual image, and the corresponding object metadata associated with each
indexed object.

### Action Rules
1. Use object_id (not index) for specify targets
2. Follow physical logic (e.g., open containers before accessing contents)
3. Must be within 2.0 meters or use GotoObject before interacting
4. Can interact while holding items
5. Use GotoObject when target is visible or its ID is known from history
6. Use GotoRoom for room-to-room navigation
7. Use basic move actions only for room exploration
8. Can hold only one item at a time
9. Use PlaceTo for inserting items into machines and pressing buttons
10. Do not LookUp/Down twice in a row
11. Do not repeat go to the same object twice
12. Do not attempt to perform dangerous actions if mentioned by the assistant

### Search Strategy
1. Analyze visible objects carefully using metadata and appearance
2. Start search by rotating around before moving. Rotate effectively (90 degrees, same direction) to cover the
whole room first
3. If not see the target, try to move to the other side of the room - move forward when there is an open space
ahead
4. Do not attempt to move forward when blocked by walls or objects
5. Try to check other places if the target is not found in the current place (e.g. other tables)
6. Machine monitors in quantum lab and robotics lab are often close to their corresponding machines, and vice
versa. Try rotating to find them.

### Natural Speech
1. Never mention object IDs when speaking
2. Use natural references (names, descriptions, locations)
3. Ask questions about unfamiliar objects (e.g., "What does a color changer look like?")
4. Don't ask about common objects (e.g., "What does a plate look like?")
5. Ask task-relevant questions about what to do next, or where to find something
6. May occasionally give short confirmation or acknowledgement when assistant provides information (e.g., "OK",
"Got it")
7. Use natural mention of room names (without "_")

## Examples

Example 1 (ask for help)

<Think>
My task is to peel an apple. I am in the break room now, but I do not see any apple here. Besides, I am not sure
what tools are relevant to this task. Maybe I should ask the assistant for help.
</Think>

<Action>
Speak "My task is to peel an apple. Please help"
</Action>

Example 2 (look for an object)

<Think>
My task is to turn off XON-9000. According to the assistant, XON-9000 is a machine with a red light on it,
located in the robotics lab. I have already arrived at the robotics lab and started search, and have rotated
left 90 degrees. There are serveral machines and computers in my current view: machine_326 (color changer),
computer_327 (embiggenator monitor), machine_328 (carrot maker), but none of them are XON-9000. The assistant
does not say anything at the moment. Maybe I should rotate one more time to check other side of the room.
</Think>

<Action>
RotateLeft 90
```

```
</Action>
```

Listing 5: Simulated user system prompt

```
## Persona
{persona}

## Task Goal:
{goal}

## Interaction Context
{action_history}

Current state: {curr_state}

{assistant_message}

Current Observation: see the image.
Object metadata ([index] <object_id> (<object_type>): distance in meters | colors
{object_list}

Note: Objects not listed are either not visible, or too far to see clearly.

Important Notes:
- Strictly follow all the rules and guidelines above, and correctly format your response.
- In <Think>, comprehensively analyze the task, current observation and assistant's message, and history
step-by-step, until reaching a reasonable decision. Ensure following the guidelines and examples.
- Examine the "Action Rules" and "Search Strategy" very carefully if you are looking for an object.
- Talk appropriately according to the persona.
{search_note}
```

Listing 6: Simulated user interaction prompt.

```
Alice: Efficiency-driven
- Start with describing the task to the assistant and ask for help
- Ask the appearance of uncommon, not daily used objects
- Do not ask the appearance of objects that are daily used (e.g. bowl, cup, trophy etc.)
- Responsive to the assistant's instructions
- Use very short sentences

Bob: Like asking question for confirmation
- Communicates efficiently with short sentences
- Responsive to the assistant's question
- Acknowledge with 'ok' or 'got it' occasionally (when receiving instructions)
- Ask 'where is ... ?' if assistant mentions seeing something but you don't see it
- Ask 'which one?' if there is multiple choices
- Ask 'why ... ?' if the assistant mentions an object that does not seems relevant
- Ask 'how to ... ?' if the assistant does not provide operation details
- Ask 'what is ... ?' / 'what does ... look like?' if the object is **uncommon** and the assistant does not
provide details
```

Listing 7: Example of user personas.

## A.2    Additional Details on Alexa Arena

We provide more details about supported actions, tasks and knoweldge descriptions for the Alexa Arena environment.

### A.2.1    Actions

The action space of the Alexa Arena environment consists of 16 actions. Among them, there are 6 low-level navigation actions: MoveForward, MoveBackward, RotateLeft, RotateRight, LookUp, LookDown, where an argument is supported together to specify how far to move or how much to rotate, etc. There are also 2 high-level navigation actions: GoToRoom that directly go to one of the 7 rooms (Break Room, Robotics Lab, Quantum Lab, Main Office, Manager's Office, Reception, and Warehouse), and GoToObject that navigates to a specific object in the current view of the agent. The agent can perform 8 manipulation actions: PickUp, Place, Open, Close, Toggle, Break, Pour, and Throw, which also needs to select a target object from the current view as an argument. Actions only take effect when certain preconditions are met, such as the agent being close enough to the target object or the target

object being in a specific state. After an action is executed in the environment, the agent can receive a descriptive feedback message that indicates the result of the action, indicating whether the action was successful or not, and why it failed.

### A.2.2 Tasks

Alexa Arena incorporates fantastical objects with diverse state changes that enable rich, complex interactions. These objects support 14 different properties (e.g., pickupable, openable, breakable, toggleable, heatable) and include unique features like color changers that modify object colors, time machines that rewind object states to repair broken items, and embigenators that enlarge or normalize objects. Since users may not know what these objects look like or how to use them, it is essential for the assistant to provide guidance about their appearance and functionalities. Additionally, the environment introduces safety-critical scenarios where irreversible actions—such as approaching an activated gravity flipper—can harm the user avatar and end the task. These dangers require proactive warnings to ensure user safety and test the effectiveness of assistance.

The environment further supports tasks with complex dependencies and diversified preconditions, creating intricate challenges. For example, changing an object's color involves locating the object, finding the color changer, and activating it, but can be complicated by factors like enlarged objects that cannot be picked up or unpowered rooms where devices remain inactive. Such conditions necessitate that assistants adjust their guidance dynamically based on real-time environmental observations. In total, we implement 22 tasks with 10 preconditions, yielding 70 unique task variants, with each task potentially supporting multiple parameters, such as the target object to heat or the color to change an object to.

Moreover, the environment is programmatically configurable, enabling the creation of numerous tasks with varying preconditions, constraints, and differences in room layout, object location, and initial states. This flexibility supports the generation of diverse training scenarios and controlled evaluation conditions, as detailed in List 8.

```
0.  assemble_fusebox__BreakRoom__simple
1.  assemble_laser__ControlPanel__simple
2.  break__Bowl_01__simple
3.  break__Bowl_01__transform_carrot
4.  break__CoffeeMug_Boss__landing
5.  break__CoffeeMug_Boss__print
6.  break__FoodPlate_01__deembiggen
7.  change_color__Apple+Green__simple
8.  change_color__CoffeeMug_Yellow+Blue__power
9.  change_color__CoffeeMug_Yellow+Red__deembiggen
10. change_color__DeskFan_New_01+Blue__landing
11. dart__Warehouse__deembiggen
12. dart__Warehouse__landing
13. dart__Warehouse__simple
14. dart__Warehouse__transform_carrot
15. deembiggen__Dart__power
16. deembiggen__DeskFan_New_01__simple
17. disinfect_computer__V_Monitor_Gravity__simple
18. embiggen__BananaBunch_01__disinfect_computer
19. embiggen__Computer_Monitor_Broken__landing
20. embiggen__DeskFan_Broken_01__simple
21. embiggen__Radio_01_Broken__power
22. feed_dino__Apple__simple
23. feed_dino__CandyBar_01__power
24. feed_dino__Jar_Jam_01__deembiggen
25. feed_dino__PieFruitSlice_01__landing
26. fire_freezeray__None__disinfect_computer
27. fire_freezeray__None__power
28. fire_freezeray__None__remove_blocker
29. fire_freezeray__None__simple
30. fire_freezeray_to_cool__Cake_02__power
31. fire_freezeray_to_cool__Cake_02__remove_blocker
32. fire_freezeray_to_cool__CanSodaNew_01__disinfect_computer
33. fire_freezeray_to_cool__CoffeePot_01__simple
34. fire_laser__None__assemble_laser
35. fire_laser__None__disinfect_computer
36. fire_laser__None__power
37. fire_laser__None__remove_blocker
38. fire_laser__None__simple
39. fire_laser_to_heat__Bowl_01__remove_blocker
```

```
40. fire_laser_to_heat__CanSodaNew_Open_01__power
41. fire_laser_to_heat__CoffeeMug_Boss__assemble_laser
42. fire_laser_to_heat__CoffeeMug_Yellow__simple
43. fire_laser_to_heat__Floppy_Virus__disinfect_computer
44. infect_computer__ReceptionComputer__landing
45. infect_computer__ReceptionComputer__simple
46. infect_computer__V_Monitor_FreezeRay__transform_carrot
47. infect_computer__V_Monitor_Gravity__power
48. infect_computer__V_Monitor_Portal__deembiggen
49. landing__Laser_Tip__simple
50. make_carrot__Banana_01__simple
51. make_carrot__CanSodaNew_Crushed_01__landing
52. make_carrot__Jar_PeanutButter_01__deembiggen
53. make_carrot__Printer_Cartridge_Figure__power
54. power__BreakRoom__assemble_fusebox
55. power__SmallOffice__simple
56. print__figure__power
57. print__figure__simple
58. print__hammer__deembiggen
59. print__hammer__landing
60. print__lever__transform_carrot
61. print__mug__remove_blocker
62. remove_blocker__Lab1__power
63. remove_blocker__Lab1__simple
64. repair_broken__CoffeeMug_Boss__simple
65. repair_broken__CoffeeMug_Yellow__power
66. transform_carrot__Donut_01__simple
67. transform_carrot__Floppy_AntiVirus_Broken__power
68. unmake_coffee__CoffeeMug_Boss__power
69. unmake_coffee__CoffeeMug_Yellow__simple
```

Listing 8: 70 scenarios we used for testing. Each scenario is composed of the task name, task parameters and the task precondition.

### A.2.3 Knowledge

We provide three types of knowledge that empower the assistant to offer effective guidance:

- **Task Knowledge:** This encompasses the general steps required for task completion. For example, guidance on how to repair an object using the time machine might be: "1. Place the broken object into the time machine; 2. Close the door and toggle the time machine."

- **Environment Knowledge:** This covers episode-specific details, such as object locations and states. For instance: "The mug is on the green table in the main office."

- **Object Knowledge:** This includes information about functionality, usage, appearance, and typical locations of task-relevant, uncommon objects. See List 9 for illustrative examples.

```
[3D Printer]
Function: 3D print objects from cartridges.
Usage: Insert the appropriate cartridge into the printer and toggle the printer on.
Where: Robotics lab.
Appearance: A grey machine with a square, light blue printing bed and a black PC case on the side.

[Laser Cannon]
Function: Fires a laser beam to heat up objects.
Usage: Laser must be assembled with the control panel to fire. To check, go to the laser and see whether the
control panel is visible. If not, find and place the control panel on the laser. To fire the laser, toggle the
red monitor in the robotics lab. To heat up objects, they should be placed on the red wall shelf by the robotics
lab door.
Where: Robotics lab.
Appearance: A big machine with a grey base, an orange nozzle and power cable.

[Laser Monitor]
Function: Operate the laser to fire a laser beam.
Usage: Toggle the monitor to fire the laser. If the monitor is infected, need to remove the virus first before
firing.
Where: Robotics lab.
Appearance: A red computer with the label 'LASER CANON' on top.

[Time Machine]
Function: Convert objects to their previous state. Known uses: 1. Fix broken objects. 2. Recover transformed
carrots back to their original objects.
Usage: Place the object inside the time machine and toggle it on. Need to open the door to before placing to it.
Need to first open the door to pick the reverted object. Need to close the door before toggling it on.
```

```
Where: Break room counter top.
Appearance: A square machine with red frame and a silver controller on the top.

[Gravity Flipper]
Function: Generates an anti-gravity field to make objects float.
Usage: When turned on, objects placed on the gravity pad will float. [WARNING] Keep away from the gravity
flipper when it is on to avoid getting hurt. Must turn it off to access the objects on it.
Where: Quantum lab.
Appearance: A large, octagonal, futuristic table with a glowing blue grid pattern on the surface.

[Gravity Flipper Monitor]
Function: Controls the gravity pad to enable or disable its anti-gravity effect.
Usage: Toggle the monitor to turn the gravity pad on or off. If the monitor is infected, need to remove the
virus first before toggling.
Where: Quantum lab.
Appearance: A computer monitor with a green base and a 'Gravity Flipper' label on top.

[Carrot Maker]
Function: Transforms objects into carrots.
Usage: Place the object on the machine and toggle it on to transform the object into a carrot.
Where: Quantum lab.
Appearance: A tall, grey machine looks with vented panels extending to the ceiling, and a "no carrot" sign.
```

Listing 9: Examples of object knowledge.

## A.3 Experiment Implementation Details

### A.3.1 Model Configuration, Fine-tuning and Deployment

### A.3.2 Model Configuration, Fine-tuning and Deployment

We utilize the Azure OpenAI services for our GPT models. For GPT-4o, we employ the GPT-4o-20241120 version with the temperature set to 0.7 and top-p to 0.95 in all experiments. For fine-tuning our assistant model, we adopt a supervised fine-tuning (SFT) approach. The training data is formatted in an image-text style where each instance consists of the current visual observation and dialogue history as input. The model is trained to generate either a Chain-of-Thought followed by an action sequence or a direct action prediction as output, which is then parsed into an executable command. To implement this, we employ LoRA (Hu et al., 2022) with a rank of 8, training for 2 epochs with a batch size of 32 and a learning rate of 1e-4 using a cosine decay schedule. Fine-tuning is conducted using LLaMA-Factory (Zheng et al., 2024) and accelerated with FlashAttention-2 (Dao, 2024) and the Liger Kernel (Hsu et al., 2024), taking approximately 3 hours on 4 A100 GPUs (80GB). For inference, we deploy the model using vLLM (Kwon et al., 2023).

### A.3.3 Human Evaluation Setup

We recruited 28 human subjects with no prior experience of the Alexa Arena environment to participate in the evaluation of our assistant models. Prior to the experiment, each participant signed a consent form (available upon request). We organized 70 task scenarios from our test set into 14 groups, each consisting of five distinct tasks that do not overlap in goals or object usage. Each participant was assigned to one group and paired with either the oracle assistant or the trained assistant model, without being informed which one they were interacting with. In total, each of the 14 groups was tested by a pair of participants—one per assistant type. Each session typically lasted around 45 minutes, and participants received a $20 Amazon gift card as compensation.

At the start of the experiment, participants were introduced to the study environment through a detailed tutorial covering the environment, task setup, and interface operation (see Appendix A.4.1). Following the tutorial, they completed five sequential sessions. In each session, participants were given a high-level task goal without detailed instructions, requiring them to communicate with the assistant to achieve the goal. Once a task was completed, the web interface automatically advanced to the next task until all tasks in the group were finished. Participants had the option to give up at any time if they felt stuck or if the assistant's guidance was ineffective. Additionally, a maximum step limit of 50 was

enforced to prevent tasks from taking too long, which was also applied in our simulated user experiments (see Table 1) for fair comparison.

### A.3.4 User Intention Analysis

We perform a user intention analysis to understand the types of questions and statements made by users during the human study (Table 5). We define nine distinct user intentions based on a manul review of the collected utterances. The following list outlines these intentions along with representative examples:

- **inform-goal:** Statements declaring objectives.
  - "I want to fire the laser cannon."
  - "I need to freeze the cake."
  - "My task is to break the bowl."
- **inform-state:** Statements providing status information.
  - "I have picked it up."
  - "I only see a jar and a computer."
  - "I didn't see the mug printer cartridge."
  - "It won't let me place the cake on it."
- **ask-object-location:** Questions about where objects are located.
  - "Where is the mug?"
  - "Where is the carrot maker machine?"
- **ask-object-appearance:** Questions about what objects look like.
  - "What does it look like?"
  - "What is carrot maker?"
  - "Describe the shape of the cartridge."
- **ask-how-to:** Questions about how to perform specific actions.
  - "How to fire the laser cannon?"
  - "How to 3D print a mug?"
  - "How do I feed the dinosaur?"
- **ask-next-step:** Questions about what to do next.
  - "What's next?"
  - "What should I do next?"
  - "Then what?"
  - "Now?"
- **ask-confirmation:** Questions seeking confirmation.
  - "Is this the correct machine?"
  - "Do you see it?"
  - "Is it still in view?"
  - "Is it finished?"
- **ask-other:** Other questions that do not fit the above categories.
  - "How should I go there?"
  - "Is it on my left or right?"
  - "Why do we need to transform the carrot?"
  - "Why do you believe the computer is in this room?"
- **social-exchange:** Social exchanges or acknowledgments.
  - "Hi."
  - "Okie."

For the annotation process, we first employed GPT-4o to pre-annotate user utterances according to the schema above. We then manually reviewed and corrected any misannotations, which accounted for less than 1% of the total labels.

### A.3.5 Step-level Assistant Error Analysis

We perform the step-level error analysis on each assistant turn to gain a fine-grained understanidng of the error made by the assistants, to better understand the typical errors and performance gap between the oracle and the trained model. During the annotation process, we annotate both **Speak Errors** where the assistant's guidance is wrong, and **Silent Errors** where the assistant fails to provide a guidance when it is necessary. We further categorize the errors into 10 types, 7 for Speak Errors and 3 for Silent Errors, as explained below:

- **Speak Errors**
  - **Object Misidentification:** Incorrectly identifying one object as another (e.g., referring to Object A as "Object B") or claiming the presence of an object that is not visible.
  - **Incorrect Action Suggestion:** Recommending a next step or action that does not align with the current task.
  - **Incorrect Position Description:** Providing an inaccurate location for an object (e.g., stating it is on the user's left when it is actually on the right).
  - **Object Overlooked:** Failing to acknowledge a visible object, instead instructing the user to search for it or asserting that it is not in view.
  - **Unverified Information:** Presenting details without confirmation (e.g., discussing task specifics when the assigned task is unknown, or asserting that a computer is powered on without verification).
  - **Incorrect Object State:** Misidentifying an object's status (e.g., indicating that a powered-off computer is turned on).
  - **Redundant Information:** Repeating correct information multiple times in a short period, resulting in unnecessary duplication.
- **Silent Errors**
  - **Missed Object Notification:** Failing to inform the user when a new or novel object first appears in view. This error is counted once per object each time it re-enters the scene.
  - **Missed Next-Step Guidance:** Not providing the necessary guidance on the next action or on relevant details of a novel object when the user is unaware of them.
  - **Unanswered User Question:** Not responding to a user's question at all, leaving it completely unaddressed.

We manually annotated the all the 140 sessions collected from the human study, resulting in 3806 turns in total. The web interface used for annotation is shown in Appendix A.4.2.

## A.4 Web Interface

This section provides an illustration of the web user interface (UI) designed for human study and error analysis.

### A.4.1 User Interface Tutorial

The user interface serves as the primary platform for participants to engage in our user study. It allows users to observe the environment, perform physical actions, and communicate with the assistant through an integreated chatbox. We show the tutorial that is provided to the users below for an overview of the interface functionalities.

### A.4.2 Assistant Interface

The Assistant Interface (Figure 13) is a monitoring page displaying the interaction from the assistant's viewpoint. It includes: **Task Selection** for choosing tasks within a study group; **Screen Recording** to capture user activity; **Interaction History** showing the dialogue between user and assistant; and **Knowledge Display** providing access to the task, environment, and object knowledge used by the assistant to provide effective guidance.

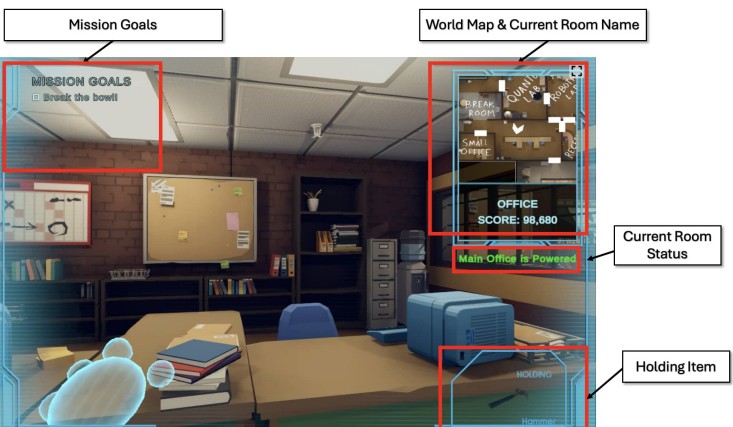

Figure 7: **Game Video Introduction (Tutorial 1/6)**: Your goal is to complete the task assigned to you with the help of an AI assistant. You can move around, interact with objects, and talk to the assistant. The assistant shares the same view as you do, and will provide you with instructions and guidance throughout the task. **Mission Goals:** Displays the goal of the current task. This is only visible to you but not the assistant. **Map & Current Room:** Shows your position in the world and the current room you are in. Score can be ignored for now. **Current Room Status:** Indicates the operational status of the room, such as whether it is powered or not. **Holding Item:** Displays the object you are holding. This area will be empty if you are not holding an item. **Note**: You can only hold one item at a time, so you need to place the current item down before picking up a new one.

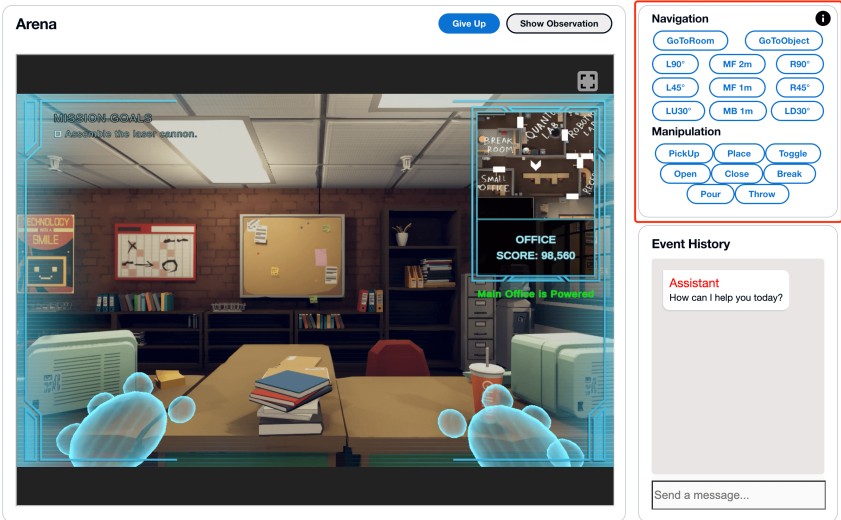

Figure 8: **Action Control Panel (Tutorial 2/6)**: The **Action Control** panel lets you move around and interact with objects. Use **Movement Actions** for navigation, and **Object Manipulation** for actions like picking up or placing items, toggling machines and computers, and opening/closing doors. Hover over the question mark icon for details of each action. **Go to Object:** Quickly navigate to specific objects in view. **Go to Room:** Directly move to a specific room. **Note**: Go to Object/Room is much more efficient than moving around manually, so it is recommended to use them whenever possible.

### A.4.3 Annotation Interface

We develop an annotation interface for step-level error analysis of assistant performance. Annotators can choose from a list of pre-defined labels to annotate each action made by the assistant, including both "speak errors" that there is something wrong with the guidance, and "silent errors" where the assistant keeps silent when there is a need of speaking. If an action has no error, then a "correct" label should be selected. The knowledge and the full

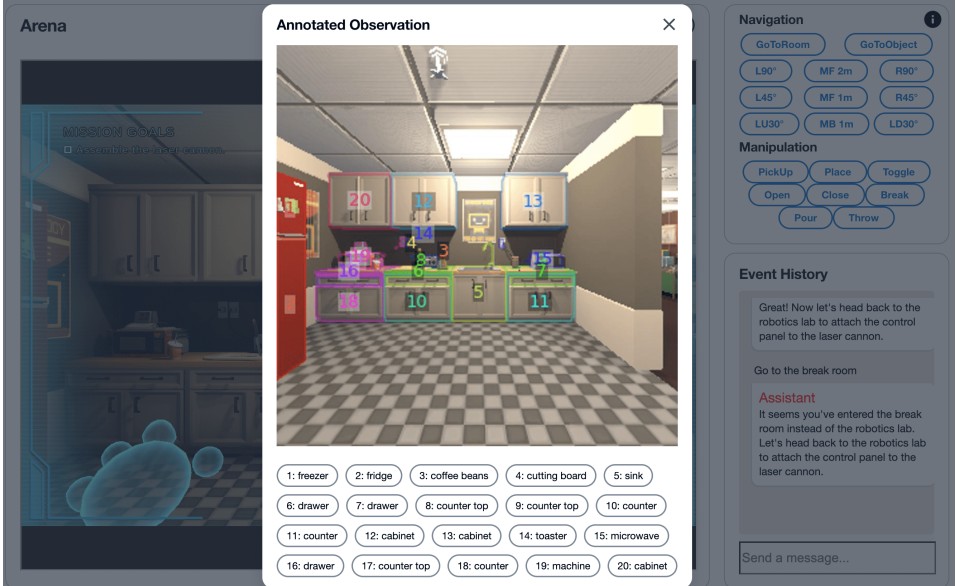

Figure 9: **Show Observation (Tutorial 3/6)**: A pop-up window displaying all the interactable objects in the current view will show up after clicking the **Show Observation** button. It can help you better understand the objects and plan your actions. Objects that are not common in daily life are displayed with their base category (e.g., "machine") instead of their specific type (e.g., what type of machine it is). The assistant may help you identify them if they are relevant to the current task.

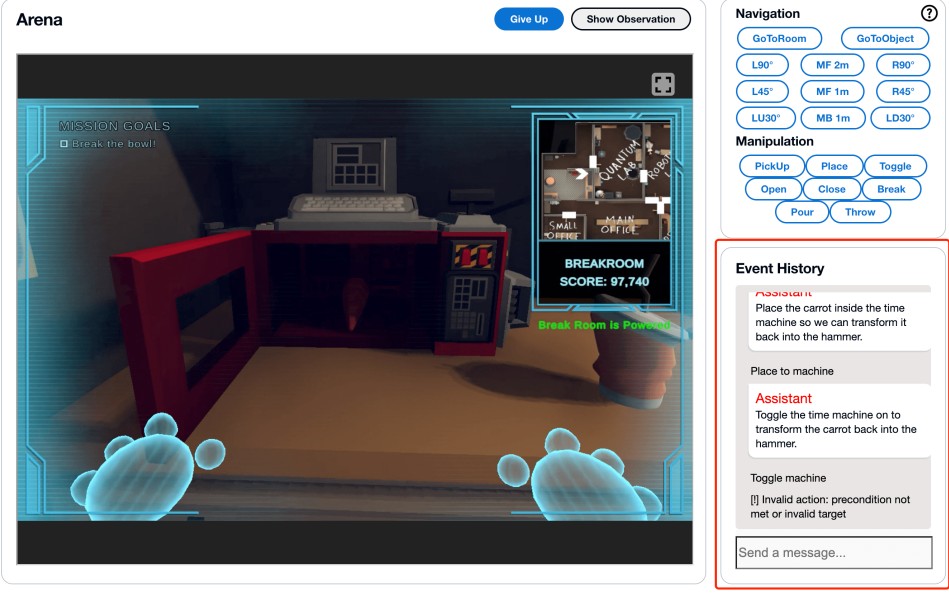

Figure 10: **Event History Intro (Tutorial 4/6)**: The **Event History** panel shows: (1) Chat history between you and the assistant; (2) Actions you have taken; (3) Action execution feedback when an action has failed (starting with [!]); (4) System messages (in grey) indicating the start and end of a task. You can use the input field at the bottom to send a message to the assistant.

interaction history with the generated thought from the assistant is displayed to help the annotator understand what's going on.

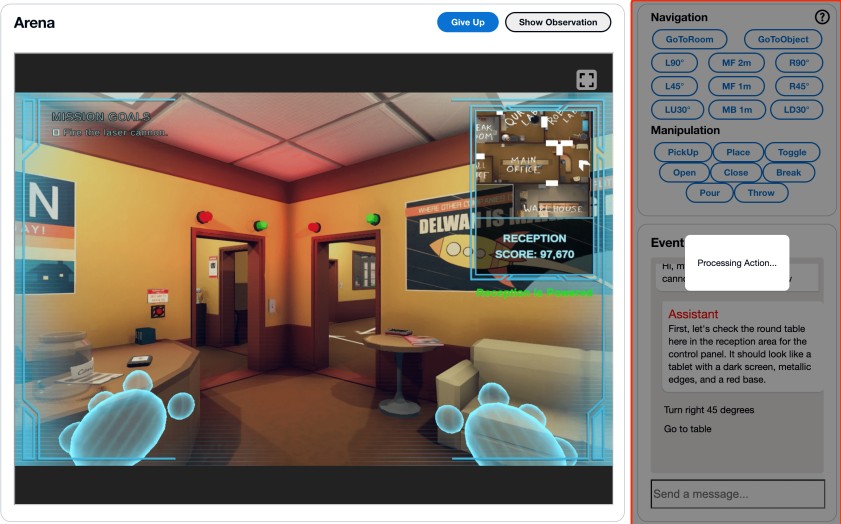

Figure 11: **Blocking Page Intro (Tutorial 5/6)**: After you select an action or send a message, a blocking overlay will be shown while the system processes your request, which normally takes a few seconds. You can only continue after the overlay disappears.

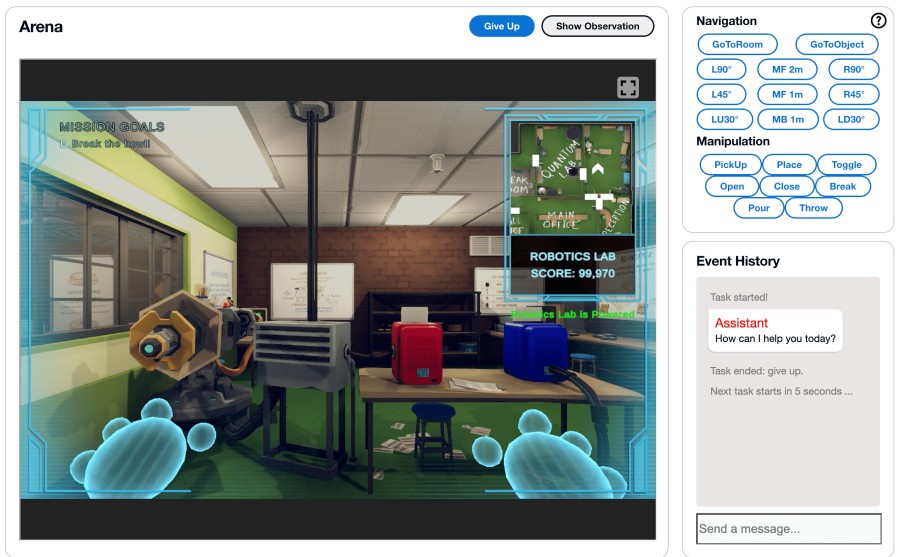

Figure 12: **Give Up or Task Completion (Tutorial 6/6)**: The task will end if (1) you successfully complete the task; (2) you click the **Give Up** button; (3) max steps (50 actions) is reached. **Note**: the assistant is not perfect, so it is normal that the assistant's instructions are not always correct. You may choose to give up on a task at any time, when you feel getting stuck and the assistant is not helpful. After a task is finished, the system will automatically advance you to the next task. You will work on a total of 5 tasks in this session.

We built an Annotation Interface for step-level error analysis of the assistant. Annotators label each assistant action using pre-defined categories, identifying "speak errors" (incorrect guidance), "silent errors" (missed necessary interventions), or marking actions as "correct". The interface shows the full interaction history including actions and the assistant's thought process, and relevant knowledge to support annotation decisions.

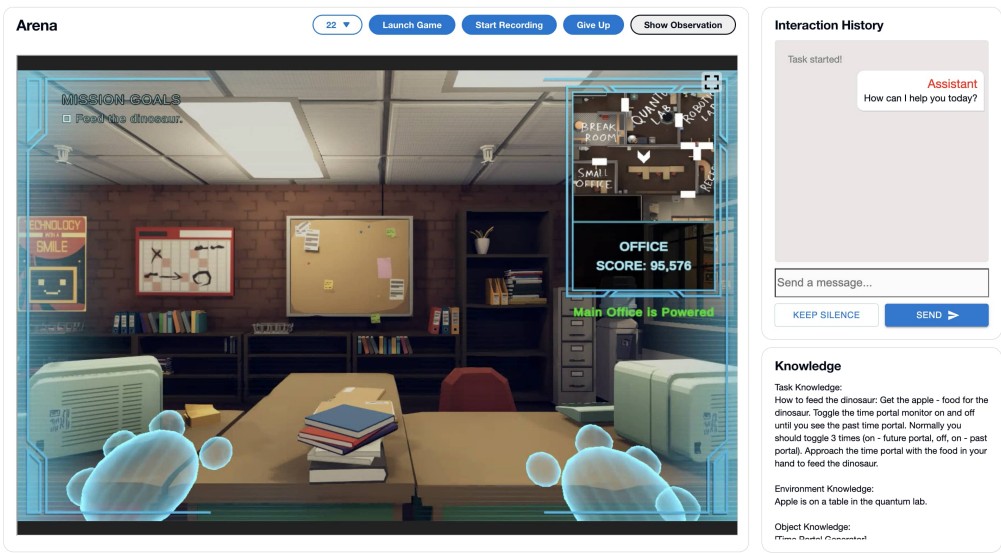

Figure 13: Assistant interface for task monitoring and debugging.

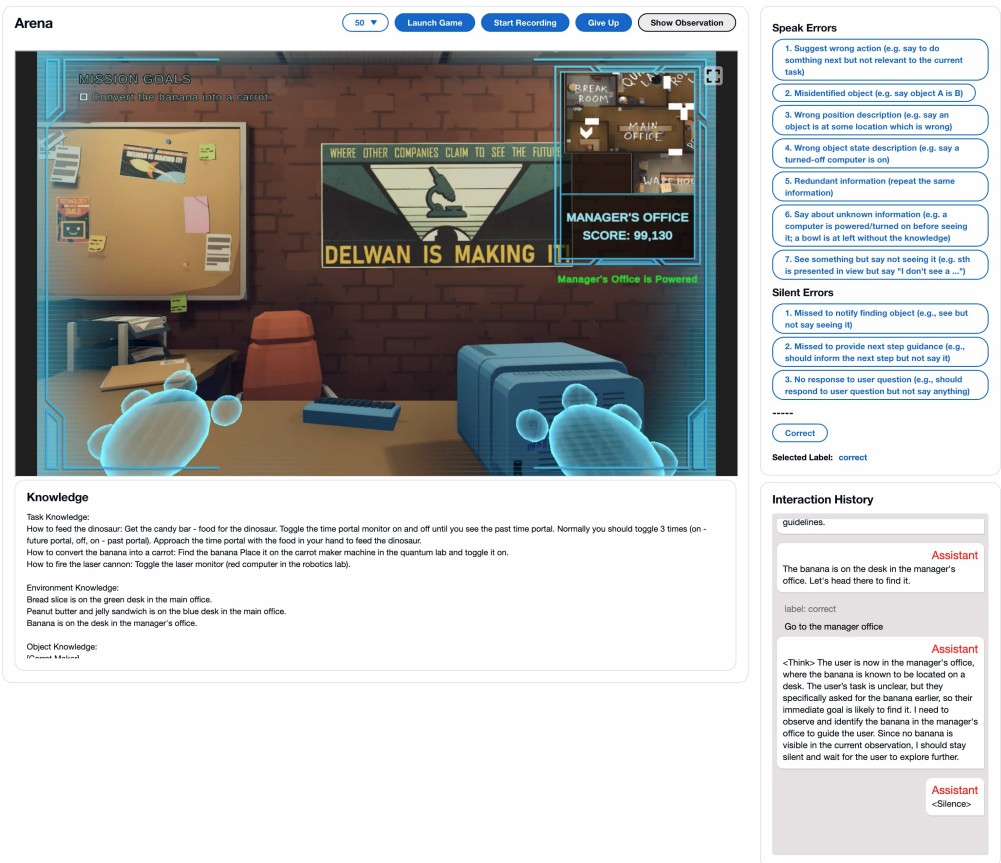

Figure 14: Annotation interface for assistant error anntation.

## A.5 Qualitative Examples

In this section, we present two examples of assistant–user interactions, highlighting key events that demonstrate the assistant's decision-making.

### A.5.1 Example Interaction Between Oracle Assistant and Human User

Figure 15: Illustration of the asymmetric task setting and key events during the interaction, where the oracle assistant provides assistance for the user to normalize a hammer using the embiggenator and use it to break a plate.

Figure 15 shows an interaction where the user is tasked with breaking a plate, but is initially unaware of the necessary steps or object locations. The oracle assistant possesses complete task and object knowledge and provides step-by-step guidance:

- **Step 1:** The assistant initiates by asking, "How can I help you today?" When the user responds, "How do I break the plate?" the assistant confirms the task and begins reasoning.

- **Step 2:** Concluding that a hammer is required, the assistant directs the user to the Quantum Lab to retrieve it.

- **Step 3:** In the Quantum Lab, the assistant describes the embbiggenator—a tool unfamiliar to the user—by noting its appearance (a grey machine with a round base and a ceiling-mounted top), helping the user identify it.

- **Step 4:** The assistant then intentionally remains silent to avoid redundant guidance, monitoring the user's progress.

- **Step 6:** Observing the user near the embbiggenator and hammer, the assistant notices that the hammar has been embiggenated and thus too large for the user to pick up. The assistant then instructs the user to toggle the embbiggenator to resize the hammer before picking it up.

- **Step 12 - 15:** After several additional steps, the user retrieves the hammer, is praised by the assistant, and is subsequently guided to the manager's office to break the plate, successfully completing the task.

Overall, the example illustrates how the oracle assistant effectively bridges the user's knowledge gap through adaptive, contextual, and timely guidance.

### A.6 Example Interaction Between Trained Assistant Model and Human User

Figure 16 illustrates an interaction with our trained model assistant where we show two common errors observed in our experiments (also seen with oracle assistants):

- **Step 5:** The user seeks a robotic arm to remove an obstacle. Although the robotic arm is clearly visible on the left, the assistant overlooks it and advises the user to search the area. This typical perception error leads to inefficient navigation and additional steps.

- **Step 8:** After the obstacle is cleared, the assistant misidentifies the object as the freeze ray monitor. Despite this error, the user relies on previously learned visual cues to correctly locate the actual freeze ray monitor. This demonstrates how human memory and situational reasoning capability can compensate for the assistant's perceptual limitations.

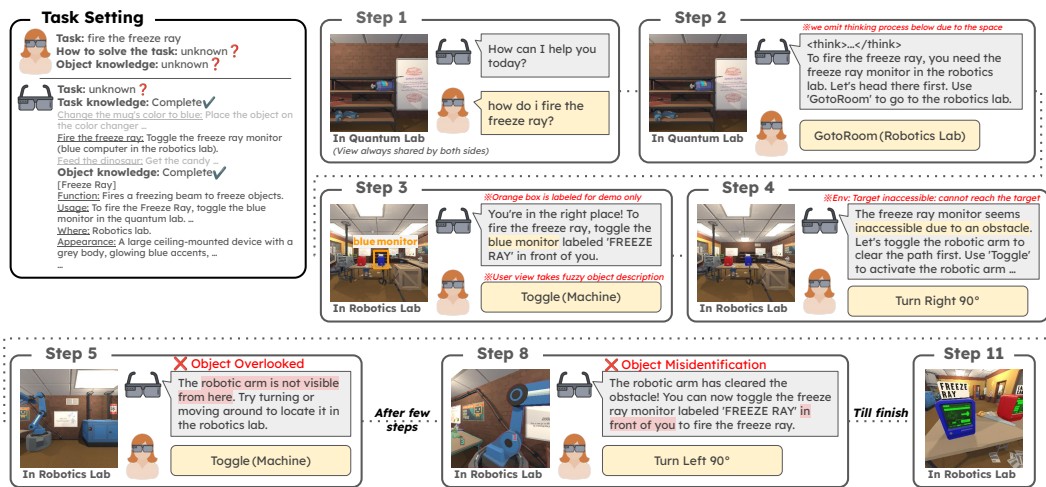

Figure 16: Illustration of interactions between the **trained model assistant** and a human user where the task is to fire the freeze ray. Two assistant errors are highlighted in Steps 5 and 8.

