# OpenReview forum: "Bootstrapping Visual Assistant Modeling with Situated Interaction Simulation"
_colmweb.org/COLM/2025/Conference — COLM 2025_

### Official Review · Reviewer_EikH · 2025-05-05

**Rating:** 7
**Confidence:** 3
**Ethics Flag:** 1

**Summary:**

The paper proposes a novel training strategy for visual assistants, that is, assistants that have access to visual and audio perception are meant to help users perform physical tasks (for example in smart glasses).
The training strategy consists of three phases:
1. construction of simulated data using a simulated environment, a simulated user and an oracle assistant with access to privileged information about the environment (location, position, properties about the objects)
2. given the simulated dialog data, training of the assistant model under the actual environment constraints (visual and language perception only); the model is tuned using the simulated user as a source of evaluation data
3. evaluation of the trained model with real human users, in the simulated environment.

Each step is thoroughly evaluated, and the method is shown to outperform various state of the art approaches, without relying on annotated data.

**Questions To Authors:**

1. The chechmark convention in Table 1 for precise/fuzzy is extremely confusing, and it took me a long time to understand that precise is the X and fuzzy is the checkmark: they are reversed in the caption, and not clarified in the main body. I would suggest changing that table.

2. SOM annotation, which is provided to the oracle assistant, is not provided to the trained assistant at step 2. Yet, it is reasonable that even in a real-world physical setting, segmentation would be applied prior to passing the image to the assistant, so it would be plausible that the SOM is available. What's the rationale for omitting and relying purely on RGB data?
3. Similar question to 1, but for depth data: I understand that pretrained MLLMs are not trained with depth data, so it does not make sense as an input for step 1, but a real-world application could well have depth or point cloud data. Would this method be able to simulate this information in order to have a more robust assistant?

**Reasons To Accept:**

Previous methods to train visual assistants rely on human interactions, which are incomplete and very expensive to acquire. The proposed method instead only relies on synthetic data, thus it can scale to larger datasets at a significantly lower costs.
Using synthetic rather than annotated data is also necessary to scale the complexity of the environment action space, thus allowing more refined simulated environment that more closely match the true purpose of visual assistant (helping in the physical world).

The authors also perform extensive evaluation of each step of the proposed pipeline, showing strong results and giving confidence of the effectiveness of the method.

**Reasons To Reject:**

While the experiments are generally comprehensive, there are a few areas that could potentially be improved:
1. The goal of Exp2 should be to select the model that more closely simulates a human user. Yet, the comparison given in Table 1 is on task success rate among simulated users, not for each simulated user against a human user on the type of utterances produced. The comparison against human data in Fig. 5 is only for two models, not all 5 in Table 1. It is unclear why a _more effective_ user simulator is also a _more faithful_ user simulator, because the simulator can take shortcuts.
2. I am uncertain that Exp4 actually fulfills the goal of RQ4, which is to show that the proposed method has better transfer ability from simulated to real _users_ and not from simulated to real _environments_. Exp4 compares a model trained (so to speak) with oracle environment information to one trained with limited information, both in interactions with human users. That's an environment transfer, not a user transfer. To evaluate that the model actually transfers, one would at least compare and analyze the performance of the same model against a simulated and a real user (ie Table 2 line 4 vs Table 3 line 2), but even better it would be to compare against a reasonable amount of human data.

---

> ### Author Response · Authors · 2025-06-02
>
> We thank the reviewer for the insightful feedback and thoughtful questions. We especially appreciate your recognition of the novelty and scalability of the BASIS framework and the thoroughness of our experimental evaluations. Below we respond to each of the raised concerns:
>
> **R1: Faithfulness of user simulators and evaluation in Exp2**
>
> This is an excellent observation. In our work, we consider a good user simulator to possess two key properties:
> - (1) Instruction-following behaviors when instructions are clear
> - (2) communication behaviors to resolve uncertainty
>
> To assess (1), we note that good instruction-following behavior can be proxied by success rate with oracle assistant (guidance from whom has been verified as effective through our real human study). A high SR@1 in this setting indicates that the simulated user can interpret and execute good instructions like humans. Simulators with very low SR (e.g., o3-mini, Gemini) typically lack the reasoning or grounding capabilities expected of human users and are excluded.
>
> To assess (2), we examine behavioral signals such as average trajectory length and number of user dialogue turns. These serve as proxies for uncertainty expression—e.g., users seeking clarification or hesitating under ambiguous object references.
>
> Ultimately, we balance both criteria to select a user simulator. For example, although the simulator with precise object references achieves slightly higher SR@1, the one using fuzzy object references better mimics humanlike ambiguity and leads to more interactive dialogues. Therefore, we selected the latter as more faithful, even if slightly less effective.
>
> While we have considered this tradeoff, we acknowledge that faithfully simulating human behavior remains a significant challenge. Future work should further investigate how to quantify the similarity between model-expressed uncertainties and those of real users, as well as cover more diverse user behavior distributions.
>
>
> **R2: Interpretation of Exp4 and goal of RQ4 (user transfer vs. environment transfer)**
>
> Thank you for pointing this out. The primary objective of RQ4 is to evaluate how well an assistant trained using our framework can assist real human users under realistic conditions. In Exp4, the first row of Table 3 presents the performance of the GPT-4o oracle assistant, which receives privileged environment information; this is not a trained model, but rather serves as a reference to establish an upper bound on achievable performance. The second row in Table 3 reports the performance of our trained assistant with real users and is not intended for system-to-system comparison, but rather to contextualize how close our approach comes to that upper bound.
>
> We appreciate the suggestion to compare the same assistant model across simulated and real users as a more direct test of user transfer. We agree, and will revise the paper to include the following comparison:
>
> |Assistant | User | SR@1 |
> | -------- | ------- | ------- |
> |Oracle assistant | Sim | 77.1% |
> |Oracle assistant | Real | 88.6% |
> |Trained assistant | Sim | 68.6% |
> |Trained assistant | Real | 72.9% |
>
> Interestingly, both the oracle and trained assistants perform better with real users than with simulated ones. This is somewhat counterintuitive, as sim-to-real transfer often entails performance degradation. We hypothesize that this is because our simulated users—though effective—are limited by their own reasoning and planning capabilities, and sometimes fail to follow instructions that real humans can easily interpret and execute. This suggests that simulated users, while useful for training, may underestimate the true potential of the assistant in real-world interactions. Based on this observation, we believe that absolute performance with real human users is a more reliable evaluation signal than comparisons between simulated and real users—unless the simulator is proven to closely mirror human behavior. We will incorporate this analysis and clarification into the revised version of the paper.

---

> > ### Author Response · Authors · 2025-06-02
> >
> > **Q1: Table 1 notation (precise vs. fuzzy)**
> >
> > We apologize for the confusion caused. In the revision, we will replace the icons with clear textual descriptions and clarify the definitions in both the caption and main text for better readability.  Thank you for pointing this out!
> >
> > **Q2: Use of SOM and rationale for relying on RGB only**
> >
> > This is a thoughtful observation. While it is true that in real-world systems a segmentation module (e.g., SOM) could be used, our choice to use raw RGB input was deliberate. Including a pre-trained segmentation model would introduce additional sources of error and complexity, which could obscure the core contribution of our framework. Our goal in this paper is to evaluate whether end-to-end assistants trained entirely via simulation can succeed without modularized visual experts. This direction aligns with the recent trend toward generalist models and offers better scalability across tasks and environments.
> >
> > **Q3: Extending the framework to support depth or point cloud input**
> >
> > We fully agree with this suggestion. Most simulators, including Alexa Arena, support depth data, and integrating this modality into the simulation and training process is natural if the model choice supports depth. One of our motivations for using simulation is precisely the ability to generate rich, multimodal sensory input at scale. As multimodal foundation models begin to support depth and other 3D input, BASIS can be extended to train assistants that leverage these additional modalities. We consider this a highly promising future direction.

---

> > ### Comment · Reviewer_EikH · 2025-06-05
> >
> > Thank you for taking the time to address my comments, and thank you also for additionally running the proposed comparison of the same model in different user conditions.

---

### Official Review · Reviewer_DrY1 · 2025-05-12

**Rating:** 6
**Confidence:** 4
**Ethics Flag:** 1

**Summary:**

This paper proposes a framework named BASIS(Bootstrapping Assistant modeling with Situated Interaction Simulation) to develop visual assistants. It does not rely on expensive human data collection, but rather leverages simulation to bootstrap capable assistants in three stages. The experiments on Alexa Arena demonstrate the effectiveness of the method, and authors leverage detailed error analysis to highlights object current primary failure mode

**Questions To Authors:**

1. How to implement the training of the assistant model? by SFT?

I will discuss with authors further during the rebuttal.

**Reasons To Accept:**

1. This paper addresses an interesting topic of developing visual assistants and it is easy to follow.  It introduces a novel framework (BASIS) that bypasses the high cost of human-in-the-loop data collection by using LLM-powered simulation to generate interactive training data and conduct automatic evaluations, addressing bottleneck in developing visual assistants.
2. Experiments across oracle-human, model-simulated user, and real-user settings, and includes fine-grained error analyses that identify key challenges
3. The experiments derives interesting some insights for future research.

**Reasons To Reject:**

1. Although this paper focuses on a novel and interesting topic, the methods (i.e., pipeline) are widely used in other agentic or reasoning-intensive tasks, where (M)LLMs interact with the environment, collect data and trains themselves[1,2,3 ... ]. So, I think it makes technical contributions limited.
2. The Alexa Arena only includes 70 tasks. It is unclear whether the fine-tuned model really generalizes well to broader scenarios. No sim-to-real environmental transfer scenarios is explored, which weakens the claim that the assistant is broadly deployable in real-world settings.
3. Although the authors conduct real-user validation, human evaluation appears limited in scope and scale. Furthermore, comparisons to other SOTA visual assistant systems or baselines (beyond the internal oracle) are minimal, making it harder to assess the absolute performance improvements.
4. It is also unclear whether "88.6% performance of an oracle assistant" is good enough for the real-world application. In other words, why 88.6% of an oracle assistant can be viewed as effective? In addition, why the performance of oracle assistant can be viewed as "*an upper bound for assistant performance",* it is merely the SOTA performance of closed-source MLLM?

[1] ReAct Meets ActRe: When Language Agents Enjoy Training Data Autonomy

[2] OS-Genesis: Automating GUI Agent Trajectory Construction via Reverse Task Synthesis

[3] NNetscape navigator: complex demonstrations for web agents without a demonstrator

---

> ### Author Response · Authors · 2025-06-02
>
> We thank the reviewer for the constructive and thoughtful comments, as well as the recognition of the novelty and clarity of our framework. Below we respond to each of the concerns raised:
>
> **R1: Limited technical novelty due to similarity with existing agentic frameworks**
>
> We would like to clarify the distinction between our work and the cited literature. While it is true that agent-environment simulation has been widely explored, the core contribution of our BASIS framework lies in the simulation of situated interaction—not only between an agent and its environment, but also in **the dialogue and coordination between a user agent and an assistant agent**. In contrast, the cited works typically involve a single agent learning by acting in the environment, often with scripted or goal-conditioned behaviors. In our case, the assistant must engage in multi-turn natural language interaction with a simulated user, reason over ambiguous goals, and guide task execution collaboratively. We argue that this user-assistant simulation, grounded in perceptual context and interaction history, is a novel contribution in the domain of training vision-language assistants.
>
> **R2: Limited generalizability and absence of environmental sim-to-real transfer**
>
> We acknowledge the limitation of evaluating within the Alexa Arena domain and agree that generalizing to broader real-world environments is an important direction. However, we would like to clarify that our goal is not to claim broad real-world deployability at this stage, but rather to demonstrate the feasibility and effectiveness of a simulation-driven framework for training multimodal assistants in a self-bootstrapping manner. As noted in Section 3.3, BASIS addresses a different type of sim-to-real challenge—**from interacting with simulated users to interacting with real humans**. This is different from the typical use of “sim-to-real transfer” (which often describes simulated environment to real environment). To avoid confusion, we will use “transferability to real users” instead. We believe this is a meaningful and under-explored transfer scenario. Our real-user validation shows that the assistant trained solely via simulated dialogue achieves 72.9% SR@1, or 82% of the oracle’s performance, demonstrating promising generalization from simulation to real human interaction. Exploring other domains and coupling with environmental sim-to-real transfer is a critical next step and will be prioritized in follow-up work.
>
> **R3: Limited human evaluation and lack of comparison to external baselines**
>
> We appreciate this concern and agree that broader human evaluation would be beneficial. However, the scale of our human evaluation reflects both our resource constraints and a broader practical reality: collecting human interaction data is expensive and time-consuming. In particular, gathering our human evaluation results requires over a total of 1,000 minutes of online interaction between human subjects and our system, where each session requires up to 50 turns of interactions from the user and assistant/environment. The focus of this work is not to present a new state-of-the-art assistant model, but to study the effectiveness of the BASIS pipeline. The design of our experiments centers around evaluating four specific research questions about simulation strategies, prompting methods, and user-agent interaction patterns. We will emphasize and clarify our objective in the revised version. Within this scope, we found the evaluation sufficient to support our claims. While our real-user evaluation is modest in scale (70 tasks), it provides diverse task coverage and reveals systematic strengths and limitations of our approach. We agree that future studies with expanded baselines and evaluation breadth would further enrich this line of work.

---

> > ### Author Response · Authors · 2025-06-02
> >
> > **R4: Whether 88.6% oracle performance is meaningful, and what it represents**
> >
> > Thank you for raising this point. We acknowledge that the reported 88.6% SR@1 is subject to sampling variability, and performance improves when the oracle is allowed multiple attempts—reaching 94.3% SR@3 (when playing with the simulated user), as shown in the last row of Table 2. The gap between SR@1 and SR@3 suggests that the oracle can generate correct guidance but lacks consistency in delivering such guidance. We anticipate that oracle performance will continue to improve as vision-language models become more accurate and stable. Since our study uses GPT-4o-20241120, the best model available at the time of submission, we expect even stronger results with newer models, which have shown significant improvements (e.g. Gemini 2.5 pro). We will clarify this point in the revised version.
> >
> > By referring to the oracle assistant as an “upper bound,” we do not mean to suggest it defines the upper limit for the problem itself, but rather the upper bound performance achievable by models trained within our framework, as the oracle is the source of simulated demonstrations used for training. Since our assistant learns from the oracle’s outputs, it is unlikely to surpass its teacher. We apologize for the confusion and will update our paper in the revision. Echoing this concern, how to increase this upper-bound performance is an exciting future direction. Promising approaches include: (1) searching and filtering oracle outputs to construct higher-quality training subsets, and (2) bootstrapping from the fine-tuned assistant itself to iteratively generate improved supervision. These strategies can potentially refine or surpass the original oracle, and we plan to explore them in future work.
> >
> > **Q1: Training details of the assistant model**
> >
> > Yes, we use a supervised fine-tuning (SFT) approach. The training data is formatted in an image-text style, where each instance includes the current visual observation and dialogue/action history as input, and either a chain-of-thought + action sequence or a direct action prediction as output. We parse the text-based action into executable ones for execution.

---

> > > ### Author Response · Authors · 2025-06-10
> > >
> > > Dear reviewer DrY1, we just wanted to follow up on this thread in case there are any points that might benefit from further discussion. We're more than happy to provide additional details if needed and would appreciate any feedback you might have.

---

### Official Review · Reviewer_3Fih · 2025-05-14

**Rating:** 5
**Confidence:** 3
**Ethics Flag:** 1

**Summary:**

This paper presents a framework for visual assistant development, consisting of three stages: situated interaction simulation, autonomous model development, and real-user validation. The authors implemented the framework by prompting and fine-tuning MLLMs using a synthetic dataset generated based on 70 unique task variants derived from 22 base tasks and 10 preconditions. The total number of successful trajectories in the dataset is 1381. On real-user validation, the fine-tuned model (Qwen2.5-VL 7B CoT) achieved a 72.9% success rate compared to the GPT-4 oracle of 88.6%.

**Reasons To Accept:**

- The simulation framework is vital for fine-tuning MLLM for the complex tasks of visual assistants.
- The experiment results and detailed error analysis are thoroughly conducted.
- Qualitative examples to show the interaction between the assistant and the user.

**Reasons To Reject:**

- The presented framework is limited to a visual assistant with specific settings. Therefore, it is unclear how to generalize the framework for other problems.
- Lack of an ablation study to show the effectiveness of the Qwen2.5-VL model trained on synthetic data compared to the base model without training
- Without a comparison between synthetic and real data. It is challenging to assess the utility of the synthetically generated data.

---

> ### Author Response · Authors · 2025-06-02
>
> We sincerely thank the reviewer for the thoughtful comments and for recognizing the contributions of our work, including the structured simulation framework, thorough experiments, and detailed qualitative analysis. We address each concern raised below:
>
> **R1: Generalizability beyond visual assistant with specific settings**
>
> We appreciate this concern. Indeed, the current implementation of BASIS is applied within the Alexa Arena environment, which offers a well-scoped domain for testing the feasibility of our approach. Our primary aim is to demonstrate how multimodal LLMs can be developed through simulated user interactions without requiring extensive real-world human data. While the assistant is situated in a specific setup, the framework itself is domain-agnostic and modular: the simulation, modeling, and evaluation stages can be extended to other environments with suitable task definitions and multimodal interfaces. As noted in our future work, we are actively exploring extensions to more complex domains such as household manipulation tasks, which would provide broader validation of the framework’s generality. We view this work as a solid initial step that showcases the framework’s strong potential to be applicable across more diverse scenarios.
>
> **R2 & R3:  Lack of baseline comparing fine-tuned vs. base model, and synthetic vs. real data**
>
> We do not include a comparison with the base Qwen2.5-VL 7B without fine-tuning because we found it had trouble following the output format. Instead, we highlight the following two comparisons that can show the effectiveness of fine-tuning on synthetic data. (see Table 2; also summarized below).
>
> |Method                                                 | SR@1
> | -------- | ------- |
> |Qwen2.5-VL 7B CoT (synthetic data)  | 68.6%
> |Qwen2.5-VL 7B CoT (real human)      | 40.0%
> |GPT-4o (no training)                            | 50.0%
>
> Specifically, we directly compare the Qwen2.5-VL 7B model fine-tuned on synthetic data with (1) the same model trained on a small set of real human dialogues and (2) the stronger GPT-4o model without any fine-tuning. These results highlight two key takeaways:
> - For R2: Training on synthetic data boosts Qwen 7B performance beyond GPT-4o’s zero-shot capability, despite GPT-4o being a stronger base model.
> - For R3: The synthetic data significantly outperforms real-human data collected for the same tasks, primarily due to its larger scale and broader coverage.
>
> These findings should support the claims in question, and we will emphasize them in our revision.

---

> > ### Author Response · Authors · 2025-06-10
> >
> > Dear reviewer 3Fih, we just wanted to follow up on this thread in case there are any points that might benefit from further discussion. We're more than happy to provide additional details if needed and would appreciate any feedback you might have.

---

### Official Review · Reviewer_PSAt · 2025-05-14

**Rating:** 6
**Confidence:** 3
**Ethics Flag:** 1

**Summary:**

This paper proposes BASIS, a three‑stage simulation pipeline that bootstraps visual assistants entirely from multimodal LLM‑driven oracle/user interactions, achieving 72.9 % real‑user success in Alexa Arena and 82 % of an oracle with privileged perception. Object identification emerges as the key failure mode.

**Questions To Authors:**

1. Given 28.9 % misidentification (Figure 6b), have you tried integrating external vision backbones or active perception to mitigate this bottleneck?

2. Since only 66 oracle‑human traces are collected (line 251), do you plan to collect larger‑scale human data to cover varied communication styles?

3. How would BASIS handle real camera noise and egocentric motion in the physical world (lines 161‑165)?

**Reasons To Accept:**

1. This work reduces human effort and interactions like in Wizard-of-Oz data collection.

2. The sim-to-real transfer is successful with 72.9 % SR@1

3. Comprehensive ablations show Chain‑of‑Thought and task knowledge benefits

**Reasons To Reject:**

1. Real human training data (66 dialogues) can be limited, which raises some doubts about whether human variability may be under‑represented.

2. Experiments in Table 2 and Table 3 only involve Qwen and GPT-4o models. More open-sourced models could be experimented on for a better demonstration of the generalizability.

3. There might be experimental baselines missing regarding prior visual‑assistant systems, which could lead to comparisons restricted to in‑domain LLM variants.

---

> ### Author Response · Authors · 2025-06-02
>
> We thank the reviewer for the thoughtful and constructive feedback. We appreciate the recognition of our contributions, including the reduction of human supervision, the success of real user experiment and the comprehensive ablation studies. Below, we address the reviewer’s concerns and questions in detail:
>
> **R1: Size of real-human training data**
>
> We agree that the amount of real human training data is limited, but this reflects both our resource constraints and a broader practical reality: collecting human interaction data is expensive and time-consuming. In our case, gathering the real human data in our experiment already required over 1,000 minutes and significant hosting effort, plus $560 compensation, which is much costlier than generating thousands of synthetic trajectories. Our goal is not to argue against the value of more diverse human data, but to demonstrate that our framework can effectively bootstrap multimodal assistants with minimal human supervision. Improving performance with larger-scale human data is orthogonal to the core contribution of this work.
>
> **R2: Only Qwen & GPT-4o evaluated**
>
> We appreciate this suggestion. Our primary goal was not to benchmark across models, but to explore how different design choices in simulation affect the performance of a single assistant model. Thus, we focused our computational budget on training and analyzing several variants of the same model architecture for the control study. That said, we did evaluate multiple SoTA models in the user simulation phase (Table 1) to compare the robustness of the simulated dialogue generation.
>
> **R3: Missing prior visual-assistant baselines**
>
> Our goal in this paper is to evaluate the BASIS simulation framework, rather than to position a new model against existing visual assistant systems. While prior agents could be adapted to our setup, many are either task-specific or trained under different assumptions (e.g., pre-scripted instructions, not end-to-end trainable). To isolate the impact of simulation design choices, we chose to keep the experimental scope focused on variants within our framework. We agree that benchmarking against broader baselines is important and will be explored in future work.
>
> **Q1: Integration of external vision backbones or active perception**
>
> Thank you for the suggestion. We have not integrated external vision backbones or active perception mechanisms in the current study, as our focus is on evaluating the simulation pipeline rather than vision model enhancements. That said, incorporating stronger vision encoders or active exploration strategies could directly improve object identification performance and is a promising direction for future work.
>
> **Q2: Plans for collecting larger-scale human data with diverse communication styles**
>
> Indeed, our current experiments are conducted within Alexa Arena, a constrained environment with relatively low variability in user communication. As a next step, we plan to extend BASIS to more complex and naturalistic domains, such as household environments (e.g., TEACh [1]), where user communication exhibits more diverse linguistic and behavioral patterns. This extension will allow us to collect richer human data at a larger scale and further validate the generalizability of our approach.
>
> **Q3: Handling real-world camera noise and egocentric motion**
>
> We agree that bridging the sim-to-real gap in perception is a critical challenge for real-world deployment. In our current study, we assume relatively stable perceptual input during online evaluation. To address the noise and variability introduced by real camera streams and egocentric motion, techniques commonly used in robotics (e.g., noise injection [2] and domain randomization [3]) can be integrated during training to improve model robustness. While this aspect is not the focus of our current work, we view it as an important and complementary direction, and plan to explore it in future research.
>
>
> **References**
>
> - [1] Padmakumar, Aishwarya, et al. "Teach: Task-driven embodied agents that chat." Proceedings of the AAAI Conference on Artificial Intelligence. Vol. 36. No. 2. 2022.
> - [2] Ouyang, Hao, et al. "Neural camera simulators." Proceedings of the IEEE/CVF conference on computer vision and pattern recognition. 2021.
> - [3] Chebotar, Yevgen, et al. "Closing the sim-to-real loop: Adapting simulation randomization with real world experience." 2019 International Conference on Robotics and Automation (ICRA). IEEE, 2019.

---

> > ### Author Response · Authors · 2025-06-10
> >
> > Dear reviewer PSAt, we just wanted to follow up on this thread in case there are any points that might benefit from further discussion. We're more than happy to provide additional details if needed and would appreciate any feedback you might have.

---

### Comment · Program_Chairs · 2025-04-03

This paper violates the page limit due to adding a limitation sections beyond the page limit. COLM does not have a special provision to allow for an additional page for the limitations section. However, due to this misunderstanding being widespread, the PCs decided to show leniency this year only. Reviewers and ACs are asked to ignore any limitation section content that is beyond the 9 page limit. Authors cannot refer reviewers to this content during the discussion period, and they are not to expect this content to be read.

---

### Comment · Area_Chair_9tfS · 2025-06-04
**Invitation to Engage in Reviewer Discussion**

Dear reviewers,

Thank you once again for your thoughtful and constructive reviews of the submitted paper. The authors have carefully considered your feedback and have now submitted their responses for clarification and further discussion.

We kindly invite you to take a moment to review the authors’ replies at your earliest convenience. Engaging in this discussion phase not only helps ensure clarity and rigor in the final work but also fosters a collaborative review process that strengthens the quality and impact of research in our community.

I greatly appreciate your continued support in upholding the standards of peer review.

---

### Decision · Program_Chairs · 2025-07-08

**Decision:**

Accept

**Comment:**

This paper addresses the compelling and significant challenge of collecting human-in-the-loop data for visual assistants via simulation. The experimental setup is comprehensive, covering oracle-human, model-simulated user, and real-user scenarios, and the subsequent analysis provides valuable insights.,

The work would be further strengthened by incorporating additional experimental evaluations, particularly through expanded real-user studies.